# Unification of Mind and Matter through Hierarchical Extension of Cognition: A New Framework for Adaptation of Living Systems

**DOI:** 10.3390/e26080660

**Published:** 2024-08-02

**Authors:** Toshiyuki Nakajima

**Affiliations:** Department of Biology, Ehime University, Matsuyama 790-0826, Japan; cognizers@yahoo.co.jp

**Keywords:** adaptation, cognition, consciousness, information, inverse causality, mind and matter

## Abstract

Living systems (LSs) must solve the problem of adapting to their environment by identifying external states and acting appropriately to maintain external relationships and internal order for survival and reproduction. This challenge is akin to the philosophical enigma of how the self can escape solipsism. In this study, a comprehensive model is developed to address the adaptation problem. LSs are composed of material entities capable of detecting their external states. This detection is conceptualized as “cognition”, a state change in relation to its external states. This study extends the concept of cognition to include three hierarchical levels of the world: physical, chemical, and semiotic cognitions, with semiotic cognition being closest to the conventional meaning of cognition. This radical extension of the cognition concept to all levels of the world provides a monistic model named the cognizers system model, in which mind and matter are unified as a single entity, the “cognizer”. During evolution, LSs invented semiotic cognition based on physical and chemical cognitions to manage the probability distribution of events that occur to them. This study proposes a theoretical model in which semiotic cognition is an adaptive process wherein the inverse causality operation produces particular internal states as symbols that signify hidden external states. This operation makes LSs aware of the external world.

## 1. Introduction

Living systems (LSs) must solve the critical problem of detecting their external and internal states and responding to them to maintain beneficial external relationships with their surroundings and generate internal order for survival and reproduction. The solution to this problem is adaptation. However, how do they achieve this? Natural selection theory does not explain adaptation; it only explains the spread of adaptations if they exist. 

How can LSs detect their external reality, whether biotic or abiotic, and respond accordingly? One might answer that organisms acquire stimuli or information from their surroundings and respond to them appropriately. This is an externalist perspective in which an external observer (scientist or experimenter) observes a given LS and its surroundings from the outside. However, any LS cannot adopt this perspective because it cannot observe itself and its surroundings from the outside. This problem is a biological version of the philosophical enigma of how the self can know the existence of external reality, that is, how it can escape solipsism. This problem has been largely ignored, and there is no consensus on solutions in the scientific community. 

Let us consider a perspective from within an LS (i.e., the internalist perspective) in which no external observer is assumed. Our question can then be more precise: How can LSs confirm the reality outside by distinguishing it from internal states? How can passive physical matter comprise an LS capable of achieving this? How can such LSs develop the ability to be aware or conscious of their surroundings? Put philosophically, how can LSs, including humans, escape “solipsism” (i.e., the philosophical doctrine that only I exist)? This solipsism problem does not arise from the perspective of an external observer but only from the perspective of the LS. Therefore, it has been ignored in mainstream biology. The problems of escaping solipsism and adapting to the environment are two sides of the same coin. This problem can be transformed into a scientifically tractable form: how can LSs produce internal states as symbols that signify external states using a reliable principle-based process, according to which they can act to survive and reproduce? (See Appendix A for further details.) 

There are three options for the fundamental (metaphysical) framework to describe the world: (1) the world is made up of matter (materialism or physicalism), (2) the world is made up of the mind (panpsychism), and (3) the world is made up of a third entity that unifies the mind and matter. Materialism (physicalism) asserts that mental processes are physical [1]. Consider the translation of a DNA sequence into a protein as an example from this perspective. A protein is synthesized according to a DNA sequence through material interactions between DNA, RNAs, enzymes, and ribosomes. This process is ontic information transmission that can proceed without any observer. Suppose a biologist observes the translation process. The biologist’s process is epistemic, not ontic (material). In this case, the materialist recognizes the epistemic process as material, in which the molecular process affects the neural systems of the biologist’s brain in a specific manner. However, another observer may have observed the material process occurring between the protein synthesis and the biologist. The materialist can again understand the entire process, including the material process and observer, as a physical (ontic) process. However, this argument leads to an infinite regression. Evidence of the material world can be obtained only through sensation and perception [2] (Chapter 6). Consequently, mind–matter dualism prevails in science, leading to significant difficulties (see Appendix B for further details). The second option, panchychism, posits that the mind or consciousness is fundamental and ubiquitous in the world [3]. According to Descartes’ skepticism, the only undoubted existence is a conscious being, while physical entities may be hypothetical. However, it remains unclear how the material world described in science can be understood from this perspective. The third option proposes the unification of the mind and matter. Russell stated in his book, ‘The stuff of which the world of our experience is composed is, in my belief, neither mind nor matter, but something more primitive than either’ [4]. This perspective offers a substantial advantage in avoiding problems associated with dualism. Josephson explored this perspective to overcome mind–matter duality by extending the traditional scientific framework [5,6]. However, an explicit model for a unified entity has yet to be developed.

This study adopted the third option in an updated version by integrating the results of previous studies. The primary goal is to provide a comprehensive framework for the adaptation of LSs to their environments using a monistic model of the world, called the “cognizers system model” (“s” for “cognizer”, denoted as CS model). In this model, mental (epistemic) and material (ontic) entities are unified as a single-type entity referred to as the “cognizer”. The CS model extends the concept of cognition to include physical movements, chemical reactions, and semiotic processes. Semiotic processes (equivalently “semiosis”) are closely related to the conventional meaning of cognition and are deeply involved in mental and living processes (see Section 4). Semiosis involves signs (or symbols), which are not restricted to those in human languages but include anything that “stands for” something else. This extension does not imply that physical and chemical entities have semiotic (or mental) properties. Instead, it highlights a general aspect of cognition common to various levels of entities, namely, “a related (relational) state change”. According to this extension, a cognizer is defined as “any entity that changes its state in relation to its external states”. Thus, this new definition of cognition encompasses the physical, chemical, and semiotic processes within a monistic framework. 

The CS model provides a general framework through which we can explore how cognition at these three levels (i.e., physical, chemical, and semiotic) operate in the adaptation of LSs to their environments, and how semiotic cognition arises from physical and chemical cognitions toward understanding the origin and evolution of consciousness. To achieve both a conceptual foundation and mathematical formalism for the model, this study addresses issues concerning cognition, information, probability, and adaptation through a thorough examination of previous research, including the author’s studies on the probability of events in LSs [7,8,9,10,11], an information theory of semiosis and mind [12,13,14,15,16], and the hierarchical structures of LSs [17]. An outline of this paper is provided in Section 2, and arguments on each specific topic are presented in the following sections.

## 2. Overview of This Study

(1)Previous works suggested that LSs are characterized as systems with the following properties: (i) creating an internal organization, (ii) establishing a relationship with external states, (iii) reproduction, (iv) formation of ecosystems, and (v) evolution. Adaptation is the core concept that concerns all these properties. In this study, adaptation is defined as the property of a system to change states to maintain a particular relationship with their environments and an internal organizational order for self-making (survival) and reproduction. Differential survival and reproduction lead to the evolution of a population of LSs. Natural selection theory explains the spread of adaptive traits rather than adaptation. However, a comprehensive theory of adaptation has yet to be established [Section 3].(2)LSs must solve the serious adaptation problem of detecting external states and relate them appropriately to their surroundings. This problem is a biological version of the philosophical enigma of how the self can know the external world and escape solipsism (Appendix A). To address this problem, the author proposes a hierarchical extension of the cognition concept. According to it, “cognition” is defined as a “related (relational) state change” occurring at three levels of the nested hierarchy: physical, chemical, and semiotic cognitions. Physical and chemical entities can detect their surroundings through related state changes; therefore, their detection capabilities are defined as cognitive capabilities at the physical or chemical level. Cognitions at these levels play an essential role in the adaptation of LSs to their environments. However, cognition at these levels does not imply that physical and chemical entities are aware of their surroundings or conscious; they do not possess this property. Awareness (consciousness) emerges at the semiotic level. Nonetheless, cognitive (i.e., detection) capabilities at the physical or chemical level are essential for making semiotic cognition possible (see (5) and Section 7 in detail). Any entity at any level that cognizes its outside (surroundings) is called a “cognizer”. In other words, cognizer refers to any “entity that changes its state in relation to external states” [Section 4].(3)Adaptation enables LSs to reduce the uncertainty of events that may occur after cognition (or action) in a manner that allows them to experience a favorable overall probability distribution throughout their lifetime to survive and reproduce. Probability, entropy, and information are vital concepts in understanding adaptation. However, they contain various meanings under the same name or mathematical representation. A comprehensive adaptation theory must coherently integrate these concepts into a theoretical framework. To this aim, adaptation is explained in terms of the probability of events using simple thought experiments, in which a player draws balls from boxes under various setups. Each box is the environment for a player, and the player experiences events such as colors and sizes of balls drawn, as well as events occurring at distant places. Probability includes various interpretations. Internal probability theory clarifies the probabilities of events occurring not to an external observer, but to a system component, such as a player drawing balls. Internal probability includes the certainty of events occurring after a given cognition (or action), and the relative frequency of events after an overall range of actions. The experiments demonstrated how cognitions occurring at the physical, chemical, and semiotic levels affect the probability of events [Section 5].(4)A mathematical formalism for the cognizers system (the “CS model”) is presented, providing explicit definitions of cognition, cognizer, system, internal/external observers, the world, the causal principle, event/state, and causation as foundations for theorizing adaptation of LSs using the CS model [Section 6].(5)For semiotic cognition to operate, LSs must measure the external states and produce symbols that signify these external states (local or global), based on which they change their effector states. A semiotic cognition model is presented based on the CS model that operates on the principle of “inverse causality (IC)”; it is not “reverse causation” or “backward causation”. The IC principle postulates that if a system (or a component of a system) changes from state *x* to *y*1 in some cases and from *x* to *y*2 in others (where *y*1 ≠ *y*2), then distinct external states must exist for *x* ⟼ *y*1 and *x* ⟼ *y*2, respectively (⟼ denotes a state change). LSs invented a measurement system based on the IC principle, a process called IC operation (or IC measurement), by which a subject LS produces symbols internally (in the form of system component states) that signify past states of the external reality hidden to the subject. By producing such symbols, a system can be aware of the external world [16], which is the core function of consciousness. “Consciousness” (or “awareness”) can thus be defined in a scientifically tractable form, as the production of internal states as symbols that signify external states, which are processed for actions. According to this definition, bacteria may be conscious (aware), although their IC measurement systems are much simpler than those used in humans [Section 7].(6)A possible scenario of a primary evolutionary process or the origin of semiotic cognition is presented, in which chemical autopoietic systems operating with chemical (i.e., non-semiotic) cognition evolutionarily develop a primitive form of semiotic cognition by IC measurement systems, similar to those observed in signal transduction in contemporary bacteria. By acquiring IC operation systems, the autopoietic metabolic system can manage the probability distribution of events to be beneficial for survival and reproduction, thereby overcoming the limitations of a primitive stage where only physical and chemical cognitions operate [Section 8].(7)In conclusion, mind and matter can be unified as cognizers in the monistic framework of the CS model. Here, physical and chemical entities as cognizers can generate a higher level of cognition (i.e., semiotic cognition) if they form a specific structure to operate inverse causality, enabling the system to produce symbols that signify hidden external states and act (physically or behaviorally) to adapt to their surroundings. This operation makes LSs aware of the external world [Section 9].

In this paper, “external reality”, “external world”, “environment”, “surroundings”, and “outside” are used interchangeably. 

## 3. What Is Life?

### 3.1. Five Properties of Living Systems (LSs)

Many definitions and characterizations of life have been proposed [18]. They vary according to researchers, considering various aspects and forms of life depending on their evolutionary stages. Based on previous studies, we can characterize five properties of LSs as material systems that distinguish them from nonliving entities:(i)Making an internal organization: LSs produce system components through metabolism and maintain organizational order in the states of these components. They incorporate material/energy resources from outside and discard waists. This process forms a unity of self-making (within a generation), separating itself from others with a boundary (i.e., self-organizing). Survival refers to the maintenance of this organization [19,20,21,22,23].(ii)Making a relationship with external states: LSs create external relationships beneficial to maintaining internal organization (i.e., survival), including obtaining resources and avoiding natural enemies and physical risks. LSs sense and respond to their environments (biotic and abiotic conditions) to maintain beneficial relationships for self-making at a high degree of certainty.(iii)Reproduction: LSs reproduce their self-making organizations, transmitting orders over generations, forming a population of LSs [24].(iv)Formation of an ecosystem: The population of LSs exists as a member of an ecosystem, where they can obtain material/energy resources (as mentioned in (i)) by maintaining beneficial relationships with others (as mentioned in (ii)), which characterize specific ecological niches (positions relative to heterospecific LSs) [25].(v)Evolution: The population of LSs evolves by generating new types of self-making systems, with some replacing old types through natural selection or drift, while others, coexisting with the old, creating diversity [24].

### 3.2. Adaptation

LSs, such as unicellular or multicellular organisms, are open systems. The adaptation of LSs is defined as the property of changing their states to maintain a particular relationship with their environments in ecosystems to maintain internal organizational order, which is an internal relationship among components, for self-making (survival) and reproduction. Differential survival and reproduction lead to the evolution of a population of LSs. Therefore, adaptation concerns all five properties mentioned above.

Organismic traits in morphology, physiology, and behavior inseparably shape adaptations to their environments. For example, the morphology and internal organization of an organism affect its behavior in relation to the environment. An organism’s digestive system (including anatomy and enzymes) and preference for food items (as an external relationship) are also inseparable in adaptation. Similarly, an organism’s metabolic systems in neurons and neural networks affect its behavior, generating its relationship with the environment. 

Adaptation includes reproductive performance. LSs perform reproductive processes within a generation at the cell or multicellular level in various forms, such as binary fission, budding, fragmentation, and production of spores/seeds or eggs (ova)/sperms. LSs possess molecular and physiological mechanisms that allocate energy and material resources to reproduction. Reproduction also requires a particular relationship with the environment, such as nutrient acquisition and the identification of oviposition sites. Organisms in sexual species need to encounter mates in animals and perform seed dispersal and pollination mediated by wind, water, or pollinators in plants.

External conditions include physical and chemical conditions (e.g., temperature, moisture, pressure, and pH), in addition to biological conditions (available resources, abundance, predators, competitors, symbiotic partners, etc.). An LS alone cannot fully determine its relationship with the environment because complete determination of the relationship requires the determination of the environmental state. For example, an encounter between prey and predators occurs based on their behavior. Similarly, enzyme-catalyzed chemical reactions are determined by the molecular movements of the substrate and enzyme molecules. The relationship between LSs, including the same and different species, and their environments, including living and non-living entities, forms an organization at a higher level, that is, an ecosystem. The external relationship, including biotic and abiotic entities, for an LS can be characterized as an ecological “niche” [26,27,28]. Organisms cannot fully determine (or construct) their niches alone; instead, niches are generated by all interactions in an ecosystem (see Section 5.5).

### 3.3. Natural Selection Theory Does Not Explain Adaptation

Evolution by natural selection is referred to as adaptive evolution, in contrast to neutral evolution. However, the theory of natural selection does not explain how LSs can adapt to their environments; it does explain the spread of adaptive traits in a population, given the variation in adaptations. Darwin [29] summarizes the theory of evolution by natural selection: “As many more individuals of each species are born than can possibly survive; and as, consequently, there is a frequently recurring struggle for existence, it follows that any being, if it vary however slightly in any manner profitable to itself, under the complex and sometimes varying conditions of life, will have a better chance of surviving, and thus be NATURALLY SELECTED. From the strong principle of inheritance, any selected variety will tend to propagate its new and modified form”. 

Based on Darwin’s theory, evolutionary biologists have developed a standard model summarized as follows: given a local environment, and if the following “conditions” are met for a given population, the population will evolve automatically as the necessary outcome of a non-random probabilistic process [30,31,32,33]. (i) Variation of organismic traits (such as physiology, morphology, and behavior) in the population, including the generation of new genetic types; (ii) organisms with different traits exhibiting “differential survival and reproduction” (i.e., natural selection); and (iii) heritability of organismic traits. The degree of survival and reproduction is called “fitness”. This model, including Darwin’s original theory, does not explain how adaptation arises from organismic traits and interactions with the environment in general terms. In the theory, adaptation is defined as the ability to survive and reproduce and is not explained in terms of other basic processes [34]. Natural selection tests adaptations, it does not produce them. In other words, natural selection contributes to the spread of adaptations in a population, serving as a base for future modifications. Nevertheless, it does not generate adaptations independently.

Fitness must be defined in terms of the potential ability (not actual data) to avoid circular (tautological) explanations; actual data provide evidence for the potential ability [35,36,37]. However, we are yet to have a general theory explaining how LSs’ traits generate the ability to survive and reproduce in a given environment. Here, adaptation is inseparable from the subject-dependent environment (as discussed in Section 5.5). 

### 3.4. Is “Information” an Answer for the Adaptation Problem?

Information has a vital role in science. In the literature, many information-theoretic accounts for adaptation can be found. For example, LSs obtain information about their surroundings and act to establish beneficial relationships with them. Information obtained from a receptor is processed through signal transduction and regulates gene expression in response to environmental changes. “Information” is sometimes reified as a third entity in addition to matter and energy without an explicit definition [38,39,40]. In this paper, “information” is not used as an explanans (a term by which we explain adaptation) to avoid the risk of a circular explanation. “Information” needs to be defined in terms of more basic concepts. In the framework of the CS model, information is defined as a “related (or relational) state change” [11,14], whether occurring in an observer in relation to an observed system (i.e., epistemic related state changes, such as “knowing”) or occurring in material components in a system, such as “pattern transmission” (e.g., hard disk’s magnetic polarity patterns altered through read/write head electromagnetic states; replication and translation of DNA). 

## 4. Hierarchical Extension of Cognition

### 4.1. Conventional Meaning of Cognition

Etymologically, “cognition” means “ability to comprehend, mental act or process of knowing”, from Latin *cognitionem* (nominative *cognitio*), meaning “a getting to know, acquaintance, knowledge” [41]. The term “cognition” has a strong anthropomorphic connotation in psychology and philosophy, encompassing knowing, thinking, reasoning, and planning. In an article where 11 researchers working on cognition in these fields defined it, it was demonstrated that the word often causes debate, and pinning down a definition of cognition is difficult and varies according to the authors [42]. Despite a variety of definitions, “information” is a key term that characterizes cognition by specifying the acquisition, processing, and storage of information. Furthermore, according to Healy [43], many authors in the study of comparative cognition use Shettleworth’s definition based on information processing: “Cognition refers to the mechanisms by which animals acquire, process, store, and act on information from the environment. These include perception, learning, memory, and decision making. The study of comparative cognition is concerned with how animals process information, starting with how information is acquired by the senses” [44]. This definition covers various levels of cognition from unicellular to multicellular organisms, including plants, invertebrates, and vertebrates. 

However, as outlined in Section 3.4, this definition has limitations, particularly when information is defined as knowledge (or knowing), which makes the definition circular. In addition, information connotes another *ontic* aspect of related state change: pattern transmission [14,45]. Therefore, a more basic model is required to address this issue. One solution is to establish a set of primitive terms that define cognition and information in a general, mathematically explicit model (see Section 3.4).

### 4.2. Cognition as a Related State Change 

The concept of “cognition” is characterized as a “related state change” within the framework of the CS model, which distinguishes it from isolated state change. When a subject system is composed of subsystems, cognition involves both external and internal relations. This notion is applied across all hierarchical levels of material entities, encompassing the physical, chemical, and semiotic domains (Section 4.3). Cognition, as a related state change at each level, shares common fundamental properties, namely discriminability and selectivity, defined by mathematical formalism (Section 6.3), each placed on a continuum. The reason for this radical extension is to avoid the cost of dualism by employing a monistic world model (see Appendix B for the costs of dualism). 

The CS model extends the concept of cognition to physical and chemical levels. “Cognition” is the determination of a succeeding state in relation to external states, which is a related state change that depends on other entities in the world. Cognition is contrasted with an “isolated state change” (Figure 1A). An isolated state change from *xi* to *xj* is represented as *xi* ⟼ *xj* (⟼ denotes a state change). A related state change (cognition) from state *xi* to *xj* occurs in response to the environment in state *ei*, denoted as *fxi*(*xi*, *ei*) = *xj* (Figure 1B), where the entity in state *xi* cognizes the environment in *ei*, changing to *xj*. Cognitions among entities create their relations by changing states in a context-dependent manner. 

“Cognition” is used as defined above in the CS model, unless otherwise mentioned. Cognition is not identical to sentience, awareness, or consciousness but conceptually includes awareness or consciousness as a special kind of cognition. For example, when we say that an electron or billiard ball is a cognizer, this does not mean it is conscious; instead, it changes from a current state to a succeeding state according to its property in a context (environment)-dependent manner. Any entity that cognizes is called a “cognizer”. The world is made of cognizers in the CS model, which is a monistic model that highlights the active nature of matter, rather than passively reacting to the surroundings.

Cognition is also represented in the same way as “acting” as a state change of a cognizer in a current condition. Acting is not “acting on” (having a particular effect on) another cognizer, such as pushing and pulling. Every cognizer acts in the current condition at every moment and is not forced or controlled by the condition. For example, a stone may move when pushed. The CS mode describes this as your hand cognizing the stone and the stone cognizing your hand (changing velocity and position).

Cognition has two aspects, discriminability and selectivity. An entity can change its current state to a different succeeding state in response to different external states. Discriminability refers to the ability to discriminate between external states. Selectivity refers to the aspect of state change, where a succeeding state is selected among possible ones (“Selection” should not be confused with natural selection). 

Inarguably, the above definition of cognition in the model cannot, by default, explain “cognition” in its conventional meaning, as used in psychology and neuroscience. The challenge remains in understanding how physical and chemical cognitions can give rise to animal and human mental or conscious processes as a unique form of cognition. The controversy regarding the term “cognition” usually arises from the lack of a clear definition in relation to other fundamental concepts within a theoretical (mathematical) model. Any scientific concept is theory dependent. Therefore, “cognition” in this paper should not be confused with “cognition” used in other contexts or frameworks. The monistic model has the potential to explain biological adaptions at various developmental stages, evolving from non-living chemical systems through primitive life forms to multicellular organisms with neural systems within a single model, as discussed in Section 7 and Section 8.

### 4.3. Cognition at Three Levels

As defined above, cognition includes state determination at various levels within the hierarchy of natural systems, which are classified into three levels: physical, chemical, and semiotic. Semiotic cognition is inherent to LSs and human-made machines; however, physical and chemical cognitions also operate in LSs. 

(1)Physical cognition

Cognition at the physical level, or “physical cognition” refers to changes in the physical state of an atom and subatomic entities in relation to the environmental (external) states, such as positions, velocities, and quantum states; here “physical” is used in the context of classical or quantum physics. This cognition occurs due to physical forces such as gravity, electromagnetic forces, and weak and strong forces within atoms. Physical cognition encompasses any state changes occurring at both subatomic and macro scales, including quantum state determination such as the collapse of wave functions. Physical cognition is discriminative and selective at a physical level. 

Many examples of physical cognition through gravity and electromagnetic fields are important components of LS adaptation. These include various physical features of LSs that enable them to relate beneficially to external physical states through physical state changes. Numerous examples of such adaptations have been documented in biomimetic studies [46]. Here are a few examples: (i) Seed dispersal: Seeds are dispersed by wind, water, gravity, animals, and birds. Many plant surfaces have hair (trichomes), which can serve different functions depending on the plant. Some hairs may function as anchors for seed dispersal by animals or wind [47,48]. Morphologies such as wings (e.g., maple samaras) facilitate adaptive dispersal using gravity [49]. (ii) The leg attachment pads: Several animals, including insects, spiders, and lizards, have leg attachment pads that allow them to adhere to various surfaces. These pads enable locomotion on vertical walls or across ceilings [50,51,52]. (iii) Aquatic movement: Many aquatic animals move through water at high speeds with minimal energy input. Drag is a significant hindrance to movement. Specialized morphology and skin surface structures help reduce friction from flowing water [53]. (iv) Fur and skin: In animals, fur and skin function as thermal insulators and protectors from injuries.

(2)Chemical cognition

Cognition at the chemical level, or “chemical cognition”, refers to changes in the molecular state and conformation of a molecule in relation to the environment, leading to chemical reactions/interactions for the self-making and reproduction of living cells (i.e., metabolism, protein synthesis from DNA, and replication of DNA). The electromagnetic field generates the major forces operating at this level of cognition, leading to covalent and noncovalent bonding (ionic bonding, hydrogen bonding, metal coordination, hydrophobic forces, van der Waals forces, etc.) to generate order (structure) or relationships among component atoms [54,55]. The van der Waals force is important for coupling between molecules of high molecular weight, whereby a molecule “recognizes” the complementary shape of another molecule [56]. Chemical cognition occurs when interacting with the external states of atoms or molecules under physical conditions, such as pressure and temperature. 

Chemical cognition is discriminative and selective at the chemical level. As an example of discriminability (or discrimination), an ion channel protein or allosteric enzyme changes its molecular structure only when a particular molecule is bound. As an example of selectivity (or selection), a protein changes its molecular structure to close (or open) its channel when a given ligand molecule is bound, rather than remaining open (or closed).

Marijuán and Navarro [57] argued that molecular recognition plays a pivotal role in generating the architecture of cell organizations. Molecular recognition plays various functional roles in the organization of living cells. They categorized molecular recognition into “identity”, “complementarity”, and “supplementarity”. Identity refers to recognition by sharing *identical* molecular properties (e.g., nucleotides/DNA, amino acids/proteins, self-organization of phospholipids in membranes, and cytoskeleton structural networks). Complementarity refers to the recognition of molecular partners with *complementary* properties (e.g., DNA/DNA pairing, RNA/ribosomes, and DNA/histones). Supplementarity refers to recognition through the capability to envelop or *supplement* molecular shapes by building a complex molecular scaffold (usually of weak bonds) around the target (e.g., enzyme active sites and protein complexes; receptors/ligands).

It is controversial whether molecular recognition is a form of “cognition”, and many are doubtful [58] (p. 165). However, in the CS model, molecular recognition is *defined* as chemical cognition (though not semiotic cognition), which satisfies the definition of cognition as a “related state change”.

(3)Semiotic cognition

Semiotic cognition, with varying degrees of sophistication, characterizes a higher form of biological adaptation to environments. It evolved from primitive unicellular systems, such as chemical autopoietic systems, which operate solely with physical and chemical cognitions. This evolution has progressed through more developed stages, such as present-day bacteria, to multicellular organisms, including humans. As a form of biological adaptation, semiotic cognition is considered to have evolved and diversified into various forms, building upon physical and chemical cognitions (see Section 7 and Section 8 for details). 

The external states of LSs may change continuously. If LSs reacted to all of these local states through physical and chemical cognitions, they cannot maintain favorable relationships with their external environment. They must respond to specific external states relevant to survival while ignoring others. Furthermore, LSs are blind to distant external states under the principle of local causation (any causal effect can occur only in the locality; see Section 6). Discrimination between the states of non-local entities through physical and chemical cognitions is impossible due to action at a distance; the non-local state *correlation* of the quantum states in entanglement is not *causation* in this context. LSs have developed semiotic cognition, which can emerge at the level of cells and multicellular organisms. 

Semiotic cognition is a related state change by semiosis. Semiosis is a process that arises from a system of physical and chemical cognizers in a specific structure (see Section 7 for details). When a bacterial cell swims toward a denser volume in sugar, it is not being pulled by the sugar molecules. Similarly, when a cat finds a mouse and starts running toward it, the cat is not pushed by photons reflected by the mouse. These state changes related to the environment are examples of semiotic cognition, constituted by physical and chemical cognitions as component processes in LSs. Hoffmeyer [59] called such a sign-mediated adaptive process “semiotic causation”, which operates through mechanisms of material efficient causation. However, semiotic causation does not clarify how semiotic processes arise from non-semiotic components such as molecules (see Section 7 for a potential solution). 

Semiotic cognition is a sign- or signal-mediated cognition generated by physical and chemical cognitions connected to a specific structure. In the CS model, “sign” and “signal” are treated as the same concept. Both terms are etymologically derived from Latin signum, meaning ‘identifying mark, token, indication, symbol, etc.’ [60].

Consider a system of physical or chemical components as cognizers. If there is a relatively high correlation between states of two entities (i.e., a high amount of mutual information in Shannon’s theory), an LS can determine the state of a distant entity (object) by cognizing a local entity (a sign or sign vehicle for the object) with certainty proportional to the degree of their correlation. 

A sign is interpreted (decoded) by a subject system, where the sign signifies a certain class of object as its meaning; meanings are subject-dependent and not objectively assigned by a meta-observer. There may be many entities outside and inside an LS with which a sign can have state correlations, some of which may be relevant and others irrelevant to the teleonomic operation of the LS. Therefore, the interpretation (meaning) of a given sign involves the choice of a specific object based on its correlation and the manner in which the LS acts to give rise to a certain relationship with the object. For example, a molecule may be released from a frog. A fly at a distance detects the molecule and flies away. In this case, the fly interprets the datum from the molecule as a “dangerous frog” (object) and flies away. Conversely, if a snake detects the molecule, it will interpret them as indicating a “delicious frog” and would approach it. Here, the datum from the molecule (sign) does not indicate a neutral frog (something cognized by an external observer) but a frog with a specific meaning for the subject (i.e., a focal cognizer), which determine the subsequent “action” (i.e., selection of a succeeding state). 

To demonstrate the distinction between physical and semiotic cognitions, consider a scenario in which a person lies down on a street. If the person *opens their eyes* in response to being shaken or spoken to, this constitutes semiotic cognition. When the *body moves* in response to a hand touch, this represents physical cognition. Physical cognition alone does not necessarily result in semiotic cognition, such as the act of opening one’s eyes, which requires a specific system structure capable of performing semiotic cognition (see Section 7 for details).

Semiotic cognition, which is irreducible to lower-level laws, is essential for living processes such as signal transduction, which regulates metabolism and gene expression within and between cells [61]. Semiotic cognition is not merely an aggregate of physical or chemical cognitions governed by natural laws at their respective levels. *Escherichia coli* can utilize lactose as an energy source when glucose is absent by producing ß-galactosidase (an enzyme for lactose decomposition), with the gene for this enzyme being transcribed in the presence of lactose. Monod [62] pointed out that the fact that lactose causes β-galactosidase production is not a necessary chemical or physical connection. He called this unnecessary, arbitrary relationship ‘gratuity’ (see Appendix C for details). 

Barbieri [18] describes such arbitrary connections as ‘convention’ or ‘code’, which function as conventional rules of correspondence between two independent molecular worlds in LSs. This leads to the concept of “evolution by natural convention” as an essential component of adaptive evolution, alongside natural selection and drift. He also proposed that the cell is a codepoietic system, capable of creating and conserving its own codes [63]. Such conventions or sign systems must be diverse in the living world, reflecting differences in their semiotic spaces [64,65].

## 5. Adaptation as Managing the Probability Distribution of Events Occurring to LSs

### 5.1. Understanding Adaptation as Managing the Probability of Events

Adaptation enables LSs to reduce the uncertainty of events following a cognition or action as a related state change. However, this reduction is not the ultimate goal. As teleonomic (i.e., goal-oriented) systems, LSs aim to achieve a “favorable probability distribution” through repeated cognitions (or actions) within their ecosystem throughout their lifetime to survive and reproduce. Concepts such as probability, entropy, and information are crucial for understanding adaptation. However, these concepts have various interpretations even when using the same name or mathematical representation. Therefore, a comprehensive theory of adaptation must be developed using these concepts.

How do organismic trials produce the “potential”, rather than the actual ability to survive and reproduce? Intuitively, most researchers agree that organismic trials can impact an organism’s ability to survive and reproduce. However, it remains unclear how and to what degree a given trait influences this ability. We need a theory for adaptation that complements the Darwinian theory. According to Darwin (Section 3.3), adaptation refers to profitable traits that confer a better “chance of surviving” (or “chance of surviving and of procreating their kind” in another part of his book) on the organism in the environment. This may transform our question into a more tractable one: how can LSs manage the probability of maintaining particular relationships with their environments and internal order that are beneficial for survival and reproduction?

The mathematical theory of probability based on Kolmogorov’s axioms [66] has inspired substantial advances in scientific research. Probability is now a common language in science. However, this theory does not present the meaning of probability. This situation creates problems when the mathematical theory is applied to scientific problems. Probability has been recognized as having a dual meaning, one aspect of which is subjective (epistemic) and the other objective (ontic) [67]. In the subjective interpretation, probability measures the degree of certainty of events occurring owing to the incompleteness of knowledge (i.e., ignorance) about the object system. The objective interpretations include the relative frequency (actual or hypothetical) of an event in a population of events and the system’s propensity (or tendency) to yield a long-run relative frequency of an event [10,68,69].

The propensity interpretation of fitness enjoys popularity among philosophers of biology [37,70,71] because fitness is independent of our knowledge. According to the propensity interpretation, relative frequency is evidence of such objective propensity to produce a certain statistical average [67,68,72]. The propensity theory interprets probability as the propensity, or tendency, of the system or situation to yield a long-run relative frequency of an event. However, it is unclear what properties of systems or their components determine this probabilistic tendency and how we can determine it without an actual measurement (relative frequency). Sober [34] pointed out the circularity of this interpretation: ‘propensity seems to be little more than a name for the probability concept we are trying to elucidate’.

The traditional interpretations, subjective or objective, described the above deal with events for external observers who observe material systems from the outside. They do not apply to events that occur to internal observers acting as players such as LSs. Therefore, an alternative framework is essential, in which epistemic observations are *objectified* as the state changes of LSs in relation to their environments. Here, LSs are described as objectified subjects (i.e., cognizers) that experience events within a system. Nakajima [7] developed a theoretical framework for this type of probability, named “internal probability”, using the cognizers system model (Section 6), in contrast to “external probability”, which refers to the probability of events experienced by an observer external to the system. There are two types of internal probability: (i)The degree of certainty of an event occurring following a specific cognition (i.e., a state change) of the environment by a focal cognizer, named “internal P_cog_”. Precisely, consider a focal state change of a cognizer in a finite length of a system’s temporal state sequence (e.g., “tossing a coin from a height of 1 m”; “moving in the left direction”) as the specific condition for subsequent events to occur; the degree of certainty is measured by the ratio of the number of a focal event type that occurred (e.g., heads up; encounter with a car) to the total number of all the events that occurred (e.g., {heads, tails}; {encounter with a car, encounter with a bike, …}) following the same cognition (state change) by a focal cognizer.(ii)The relative frequency of an event occurring following an overall range of cognitions (i.e., including all types of cognition, state changes, without specifying any particular one) by a focal cognizer during the system process, named “internal P_overall_”. Specifically, consider all types of state changes of a focal cognizer occurring over a finite length of a system’s temporal state sequence (e.g., {tossing from heights of 1 m, 2 m, … and 10 m}; {moving in the left, right, and straight}) as the overall conditions for subsequent events to occur. The relative frequency is measured by the ratio of the number of events of a focal type (e.g., heads, encounter with a car) to the total number of all types of events that occurred in the sequence (e.g., {heads, tails}; {encounter with a car, encounter with a bike, …}).

The concepts of P_cog_ and P_overall_ also apply to external observers, in which case they are referred to as “external P_cog_” and “external P_overall_”, respectively [11]. Whether internal or external, P_cog_ and P_overall_ are not merely descriptions of actual ratios but reflect the properties of the focal cognizer and its environment (i.e., the properties of the cognizers system) [7,8,9,10], which will be illustrated using thought experiments in the following section.

### 5.2. Thought Experiments for Understanding Semiotic Cognition as Adaptation

Let us illustrate how the internal probability concept is involved in biological adaptation using a series of thought experiments (*A* to *E*). In Experiment A to D, a person draws one ball from a box containing ten balls, observes the ball, returns it, and mixes it up. In Experiment E, conditions are hidden from us. The trial was repeated (*n* trials). Here, the person is a player and simultaneously an observer (internal observer), an LS that performs semiotic cognition in these experiments. The balls and box represent non-living cognizers who change their states through physical or chemical cognitions (Figure 2).

#### 5.2.1. Experiment A

The box is opaque and contains ten balls of the same size and shape but in different colors: two orange, three red, and five blue balls (Figure 2A). What is the probability of drawing a red ball? According to the classical theory of probability, the probability of an event is given by the ratio of the number of favorable cases to the total number of equally probable cases. In this case, the probability of drawing a red ball is 3/10. 

In this experiment, the player had the same cognition about the states of the ball-containing box due to the opaque wall. The player moved their hand without adjusting specifically to the ball configuration in the box. The movements may have varied due to differences in the initial conditions of each trial. However, the conditions were random (i.e., they did not correspond discriminatively and selectively to the ball positions). Therefore, both the internal P_cog_ and P_overall_ for drawing a red ball are considered 3/10. (The actual relative frequency may vary depending on how many times (*n*) the experiment is repeated. As *n* increases, it will approach the value of 3/10 according to the law of the large number.)

What about the external P_cog_ and P_overall_ of drawing a red ball? The external P_cog_ depends on the cognition of an external observer. For example, consider an external observer such as a Laplacian demon who can perfectly discriminate between all the states occurring in the system (the box, balls, and the player) and knows the laws governing the system’s state changes. For the demon, there is no uncertainty about what will happen next. Given any cognition of the demon, the probability (external P_cog_) of drawing a red ball would be either one or zero. However, the relative frequency (external P_overall_) of the drawing a red ball is 3/10. This is because a demon exists outside the system and cannot alter the system behavior. For external observers with limited cognitive ability, the external P_cog_ may vary depending on their ability, while the external P_overall_ may be the same (3/10) provided that the external observer can identify the ball color.

#### 5.2.2. Experiment B

In this experiment, the box is transparent, and all other conditions are the same as those in Experiment *A* (Figure 2B). What are the internal probabilities (P_cog_ and P_overall_) of drawing a red ball in this case? Attention must be paid to the properties of the players. Here, players can see the colors and positions of the balls through the transparent wall, and the balls are assumed not to escape their hands. Therefore, the player can choose any color they prefer. If the player prefers red, then the internal P_cog_ for red will be 1; if they dislike red, then the P_cog_ will be zero. The degree of certainty of the event (i.e., drawing a red ball) is 1 if the handling of balls according to the position data is accurate. The relative frequency (internal P_overall_) is also 1 for red balls across all trials, provided that every cognition is performed with perfect discrimination of different ball configurations and appropriate actions during the trials. 

Cognition (including action) implies selecting a succeeding state for the player from among many possibilities. Here, “selection” does not refer to choosing a specific ball or color but rather to determining a “particular succeeding state” by actualizing one of many potential states. The property of selection, or selectivity, may vary depending on the players involved in the experiment. Players influence both the internal and external P_overall_.

Consider a modified version of this experiment using balls that vary in size, weight, or surface texture, all within the same transparent box. Suppose red balls are heavy or have a slippery surface. The internal P_cog_ and P_overall_ of drawing red might be less than 1, even if the player prefers red, leading to an increased probability of drawing the second-best color. These characteristics of balls may affect how they respond to the player’s actions (hand/finger movements for grasping and handling the balls). In other words, the balls are not passive entities but rather active participants (cognizers) that perform physical or chemical cognitions, as described in Section 4. 

Similarly, we could use mice, which can perform semiotic cognitions, instead of balls for this experiment. Mice may vary in cognitive and behavioral traits, with some being cautious and others not. In this case, a careless or slow-moving type might be drawn with a higher probability than a cautious or fast-moving one. Even under transparent conditions, the player cannot necessarily choose the most preferable mouse with a probability of 1. 

Players cannot necessarily control balls. In other words, a player cannot fully determine the relation with balls through cognition involving an action (represented by a subset of the direct product of their state sets) because a relation (subset) is established once the player and the balls have selected their succeeding states (see Section 5.4 and Section 6.3 for details). 

#### 5.2.3. Experiment C

In this experiment, the box wall is semi-transparent with pink-colored walls; otherwise, the conditions are the same as in Experiment *A* (Figure 2C). This semi-transparency reduces the player’s ability to discriminate between the different ball states (positions). What is the probability of drawing a red ball? In this experiment, even if the player prefers red balls and acts to achieve it, they might mistakenly choose an orange or blue ball because of inaccurate color cognition. As a result, the internal P_cog_ and P_overall_ of drawing a red ball will be less than one but higher than 3/10.

Acting in a discriminating manner, i.e., selecting different succeeding states in response to different environmental states (the balls’ positions in the box), is called “discrimination”. The ability to discriminate between different environment states in a given situation is called “discriminability”, a property of entities (cognizers) such as the player, balls, and the box. As illustrated in the modified version of Experiment *B*, the properties of balls or mice can also affect the probability (internal P_cog_ or P_overall_) of drawing a given color in Experiment *C*.

#### 5.2.4. Experiment D

In Experiments *B* and *C*, the player preferred red balls over other colors. Body state changes are physical events involving the muscles and bone structure of the hand and fingers. This behavior occurs through semiotic cognition, where “red” may have a certain meaning, such as “beneficial”. In other words, red balls do not exert a physical force on the player’s body to behave that way; it is similar to how people escape when feeling a “bad smell” without being physically pushed.

Let us introduce another experiment, *D*, to clarify how semiotic cognition affects the probability of events (Figure 2D). The box wall is transparent, like in Experiment B, containing 10 balls of the same color but numbered (1 to 10); otherwise, it is the same as in Experiment A. What is the probability (internal P_cog_) of drawing “ball 1”? Suppose each number corresponds to gifts, e.g., 1: a bike, 2: a car, 3: a candy… Numbers are signs for hidden (distant) entities. The answer depends on the player’s knowledge of the links between numbers and these gift items. If a player fully knows the links, the situation is similar to that in Experiment B. If a player wants a car and knows this link, they will behave selectively to draw ball No. 2 with a probability of 1; therefore, the probability of ball No. 1 will be 0. If a player knows nothing about the links, this case resembles Experiment *A*. In general, the player’s ability to interpret signs is intermediate, similar to Experiment *C*.

The degree of knowledge regarding the links may vary from entirely unknown to completely known, similar to box wall transparency in Experiments *B* and *C*. The degree of knowledge about how local situations (i.e., ball configurations) link with things (gifts) outside the box, i.e., non-local correlations, may be called “semiotic transparency”. Physical and chemical cognitions can discriminate between local states only under the local causation. However, semiotic cognition enables cognizers to discriminate between non-local states mediated through local–global state correlations.

#### 5.2.5. Experiment E

Experiments *A* to *D* share the framework of an external perspective, called the “externalist framework”, in which all components, such as balls, box, and players, are provided, and we (the reader and the author) observe the systems from a meta-level (Section 6.1). However, any LS as an internal system component cannot cognize itself and the environment from the outside. Experiment *E* provides a first-person framework, called the “internalist framework”, where only the player is situated within this framework (Figure 2E). No entities besides the subject (player) are assumed (but not denied). We suspend judgment on whether something exists external to the subject to discover how the subject can cognize external reality. Here, we see LSs from the inside out (see [73] for a similar approach in brain science).

In this case, what is the probability of drawing a red ball? It may seem impossible to answer this question. However, it is not. The player will experience a temporal sequence of data (senses and percepts) from repeated trials (the data occur within the player, not in an external observer). These include, for example, a visual image of a black box, a transparent or semi-transparent box, colored balls, mice, or numbered balls appearing in a box. Some players might experience five red and five blue balls in a box, while others might experience only one object. Note that “data” are phenomena, which do not necessarily imply that the external reality caused them. “A red ball” might be an “icon” for realizing the external reality for a player (see Section 7.4 for a related discussion). Furthermore, the senses and percepts might have occurred in dreams. The player moved the hand and fingers in response to each perceptual image. Therefore, the answer depends on the data sequence given to the player. The relative frequency of percepts of “a red ball” to the total number of “a ball” indicates the probability (internal Poverall) of drawing a red ball. The ratio of the number of percepts of “a red ball” to the total number of “a ball” occurring under the same perceptual condition is the probability of certainty (internal P_cog_) of a red ball. 

Bayesian theorem can derive these probabilities by formulating hypotheses (i.e., generative models) that describe how frequently external worlds are likely to produce events with particular distributions. In this scheme, probabilities are calculated using limited options for hypotheses, which are arbitrarily chosen. However, the probabilities in the generative models are descriptive and are not derived from a mechanistic model of interactions with a hypothetical environment (as illustrated in Experiments *B*, *C*, and *D*). Furthermore, the data may not necessarily result from external reality, which might occur in dreams. Therefore, our challenge arising from Experiment *E* is how LSs can escape solipsism and produce symbols that signify external states and process them to act reliably (Appendix A). This problem is addressed in Section 7.

### 5.3. Probability, Entropy, and the Amount of Information

The concepts of probability, entropy, and information are essential for exploring the general principles underlying the properties of LSs and for integrating various disciplines in biology. Kolmogorov’s axiomatization [66] provides mathematical consistency for the theory. However, probability encompasses a variety of interpretations, which can create problems when applying the mathematical theory to scientific issues [10,74]. According to the argument in Section 5.1, we have four types of probability: internal P_cog_, internal P_overall_, external P_cog_, and external P_overall_ (Table 1). (The mathematical relationship between P_cog_ and P_overall_ in the CS model is presented by Nakajima) [11]. 

Notions of internal and external probabilities naturally extend to entropy and the amount of information, both of which rely on the concept of probability (Table 1). Entropy (H) is derived from a probability distribution: H = ∑ P*i* log_2_ P*i*; “*i*” indicates event *i*. Accordingly, depending on the type of probability, four types of entropy (H) can be distinguished: internal H_cog_, internal H_overall_, external H_cog_, and external H_overall_. Internal H_cog_ measures the uncertainty of events occurring under a given cognition. Internal H_overall_ measures the uncertainty of the subject-dependent environment, which is interpreted as the entropy of umwelt (Section 5.5) [75]. Similarly, the amount of information (I) (not to be confused with “information” alone) is given by I = H_before_ − H_after_, where H_before_ and H_after_ are entropy values before and after receiving the datum (calculated from P_cog_ distributions) or before and after interactions among system components (calculated from P_overall_ distributions) of events or states (Appendix D for details). Accordingly, depending on the four types of entropy, the amount of information (I) includes internal I_cog_, internal I_overall_, external I_cog_, and external I_overall_.

### 5.4. Cognizers vs. Demons

Maxwell [76] devised a famous thought experiment to generate order using a vessel divided into two portions (*A* and *B*) and a “being” (called Maxwell’s demon). There is a small hole in the dividing wall, and the being can see the individual molecules, opening and closing this hole to allow only the swifter molecules to pass from *A* to *B* and only the slower ones pass from *B* to *A*. “Without expenditure of work”, the being will thus raise the temperature of *B* and lower that of *A*, contradicting the second law of thermodynamics. The being is similar to the player in Experiment *B* (Section 5.2). However, unlike the demon, organisms are open systems that work thermodynamically with varying properties.

The CS model conceptualizes material entities as cognizers characterized by specific discriminability and selectivity. Therefore, we can interpret the demon not as a physical entity but as a perspective shift, asking what occurs if molecules are endowed with certain selective and discriminative properties and interact with one another, unlike molecules in an ideal gas. Molecules can be seen as cognizers with varying properties (or abilities). They are not conscious but rather undergo related state changes based on their physical or chemical properties. Furthermore, unlike the human players in the thought experiments (Section 5.2), atomic or molecular cognitions are not semiotic. Monod [62] states that ‘it is by virtue of their capacity to form, with other molecules, stereospecific and noncovalent complexes that proteins exercise their “demoniacal” function’ (see [77] for a related argument). 

### 5.5. Environment and Subject-Dependent Environment

Using the previous thought experiments, we can distinguish between the environment and the subject-dependent environment. In each experiment, a box containing balls represents the environment, which remains the same for any player with the same box. In contrast, we can recognize another concept of the surroundings. Different players may have varying preferences for color, finger size, morphology, and behavioral properties. According to players’ characteristics and properties, such as discriminability and selectivity, the probability distributions of events (based on internal P_cog_ or P_overall_) may vary. This event set with a particular probability distribution characterizes the subject-dependent environment. This aspect of surroundings has been conceptualized as “umwelt” (plural: umwelten) by Jakob von Uexküll [78], who distinguishes umwelt from “umgebung”, similar to “environment” in the above sense. For example, in Experiment C, one player prefers red and the other orange with similar discrimination ability, the probability distribution of events must differ for them, although they share the same “environment”. Similarly, in Experiment *D*, they will experience different probability distributions if they have different semiotic actions based on different codes (Section 4.3).

Uexküll developed the concept of the function-circle (or functional circle): ‘Every animal is a subject, which, in virtue of the structure peculiar to it, selects stimuli from the general influences of the outer world, and to these it responds in a certain way. These responses, in their turn, consist of certain effects on the outer world, and these again influence the stimuli. In this way, there arises a self-contained periodic cycle, which we may call the function-circle of the animal’ [78,79]. This function-circle process involving the sensing and acting loop produces the surrounding world, called an “umwelt” for the organism, in which the subject experiences a particular set of events with a particular distribution. Sagan [80] points out that ‘Natural selection is an editor, not a creator’. Natural selection tests adaptations, leaving some as the basis for additional adaptations.

The “subject-dependent environment characterized by an event set with a particular probability distribution” can be understood as a particular distribution of encounter probabilities with various external entities, such as physical and chemical things, prey, predators, parasites, symbionts, etc. [8,9]. This distribution characterizes a relational position with other entities in an ecosystem, that is, an ecological niche [26,27,28]. The niche describes the position of an organism relative to abiotic (physical and chemical) entities and organisms of the same and different species, from an externalist perspective (i.e., from the ecologist’s viewpoint). Umwelt describes the position from the internalist perspective, focusing on how it arises from the semiotic property of a subject organism.

Niche construction theory [81] claims that organisms are active agents that construct their environment to be suitable for their survival and reproduction, similar to a thermodynamic engine working to produce less entropic states by consuming free energy. This claim is partly agreeable. However, as demonstrated in the thought experiments (Section 5.2), all entities in the world actively determine their states based on their properties. The CS model suggests that organisms cannot fully construct niches; instead, they participate in constructing their niches. In other words, niches are *generated* by inter-cognition among living or non-living cognizers in an ecosystem. For example, beavers build dams and lodges for survival and reproduction, such as protection from predators, holding food, and mating. A beaver dam or lodge depends on the existence of sediment, rocks, sticks, and so on [82]. The dam is an outcome of the entire ecosystem process and is not solely due to the beaver. 

Concerning a place in the food web in an ecosystem, Elton describes that ‘Animals have all manner of external factors acting upon them—chemical, physical, and biotic—and the “niche” of an animal means its place in the biotic environment, its relations to food and enemies’ [26] (Chapter 5). Suppose that organisms of species *Y* eat species *X* as prey; they (*Y*) are also eaten by organisms of species *Z*, forming a food relation: *X*-*Y*-*Z*. The position of *Y* in the food relation characterized one aspect of *Y*’s niche. Such a position is not constructed by the focal organisms (*Y*) but is instead generated by inter-cognitions of ecosystem components; they all participate in shaping a food web structure. 

## 6. Cognizers in the World

### 6.1. Overview

An explicit mathematical formalism is essential to avoid the risk of replacing what we try to explain with another vague concept that requires explication. The meta-observer (model constructor) describes a world that is not a member of the world, standing nowhere and nowhen (Figure 3). The world comprises cognizers. The world is closed, but systems within the world are open to the outside (if there is only one cognizer, it is the world itself). The CS model posits that the world is deterministic based on the causal principle (see Section 6.4 for the definition). Cognizers are entities that cognize other cognizers by changing their states over time, according to their properties. A “cognizers system” (“s” for cognizer) can be hierarchically organized in a nested fashion, where the parts constitute a whole synchronically or diachronically [17] (see Section 6.8). A single cognizer (e.g., a multicellular organism) can be described as a system of “sub-cognizers” (such as cells), which is therefore called a “cognizer system” (no “s” for cognizer) when the cognizer is focused on as a system of sub-cognizers when the cognizer is described in relation to the sub-cognizer. In this case, the cognizer system may have other cognizers in the environment. Cognition is a related state change characterized by the hierarchical level of a cognizer, such as physical, chemical, and semiotic cognitions (Section 4). 

The meta-observer can demarcate a subset of cognizers as a “system” (called a “cognizers system”), whose members are chosen by the degree of cohesiveness due to inter-discriminative cognitions between cognizers (Systems 1, 2, and 3 in Figure 3). “External observers” (LSs, not restricted to humans) are cognizers that perform semitotic cognitions of target systems from the outside. By definition, the influence of external observers on an observed system is irrelevant or negligible. For example, experimenters in classical mechanics are treated as external observers. Experimenters in science usually have a dual nature: they are both external observers and, simultaneously, the meta-observer of a theoretical model. A cognizers system can be modeled as a nested hierarchical system. In Figure 3, cognizers at the physical or chemical level are shown as the dots and gray circles, respectively. Cognizers that can perform semiotic cognition, such as LSs, are represented by blue circles (see Section 4.3). In this framework, “internal observers” (the blue circles in Figure 3) can be defined as cognizers who perform semiotic cognition within a system, provided that “observation” is defined as semiotic cognition, which may be referred to as a narrow definition of observation. However, we can also define “observation” as physical, chemical, and semiotic cognitions, whereby “observation” is equal to “cognition”, as defined in the CS model. According to this definition, any physical or chemical entity can be considered an observer (i.e., a broad definition of observation). In this case, the observers observe “their environments” that interact with themselves (e.g., experimenters in quantum mechanics, LSs in an ecosystem). Note that all entities within and without systems are cognizers, that is, physical, chemical, or semiotic cognizers, depending on their hierarchical structure.

To illustrate the above, Experiments *A* to *D* in Section 5 can be described as a cognizers system (e.g., system 1 in Figure 3) composed of a player (“internal observer 1”), balls, and a box. Other entities, such as molecules in the air, can be included in or excluded from the system, depending on whether they affect the behavior of the players and the balls. An external observer cognizes them through semiotic cognition without influencing the play. 

In the CS model, the meta-observer defines the world as deterministic. Therefore, the temporal sequence of the world must ensure that the current state determines only one succeeding state, according to the rule of the world. However, it is important to note that cognizers systems are not necessarily deterministic because they interact with the external world. In other words, any material system observed and described in science may be indeterministic because it is part of the world. A cognizers system is deterministic only if it encompasses the entire world. For example, a deterministic system can be defined by including only two cognizers: cognizer *A* and the rest of the world (i.e., its environment). The primary reason for employing the causal principle in the model is that if the world did not adhere to this principle, parts of the world (such as LSs, including humans) would lose the inverse causal principle (i.e., the contraposition of the causal principle) by which they realize (make it real) their external worlds (see Section 7). The hierarchical extension of “cognition” to all levels of the material world provides a monistic view that only cognizers exist. Cognizers are models for real entities constructed by a meta-observer, just as elementary particles, quantum fields, and matter in Newtonian mechanics are all models of reality developed and modified in science. In his book *Protobiology*, Matsuno asserted, ‘Although it is difficult to identify what measurement is all about because of its inevitable anthropocentric connotation, a common denominator is that measurement is realization of a particular pattern of interaction between an arbitrary pair of interacting bodies. Measurement is thus ubiquitous in any system of interacting bodies. What is more, every interacting atom and molecule can serve as an internal agent of measuring others interacting with it’ [83] (p. 53). Matsuno’s “measurement” may align with “cognition” as defined in the CS model.

### 6.2. Cognizers System (CS) Model

Consider a simple case in which the world comprises only one system with two cognizers, *C*1 and *C*2, with state sets **C1** and **C2**. Cognizers’ names are denoted in italics, with state sets in boldface. *C*2 is the environment of *C*1 and vice versa. 

As illustrated in Figure 4A, the temporal state sequences of *C*1 and *C*2 are given, respectively, as follows:*c*10, *c*11, *c*12, …,
*c*20, *c*21, *c*22, ….
The entire state sequence is given as follows
(*c*10, *c*20), (*c*11, *c*21), (*c*12, *c*22), …,(1)
where *c*1*i* ∈ **C1**; *c*2*i* ∈ **C2**; (*c*1*i*, *c*2*i*) ∈ **C1** × **C2**.

In the model, temporal sequences are not mere chronicles but are determined by cognitions between cognizers. Cognition is defined as the determination of a succeeding state from the current state according to a deterministic rule called a “cognition function”. (This function was originally called “motion function” in previous studies published by the author; however, “cognition function” is used in this paper for consistency).

*C*1 (in *c*10) cognizes *C*2 (in *c*20), changing to state *c*11. Similarly, *C*2 (in *c*20) cognizes *C*1 (in *c*10), changing to state *c*21, as shown in Figure 4B. The state changes in cognizers are determined by their cognition functions, *f*1 and *f*2 for *C*1 and *C*2, respectively, where *f*1: **C1** × **C2** ➝ **C1**; *f*2: **C1** × **C2** ➝ **C2** (“×” denotes the direct product of sets; “➝” denotes mapping). *f*1(*c*10, *c*20) = *c*11, *f*2(*c*10, *c*20) = *c*21. Each cognizer (*Ci*) is identified with the cognition function, *fi*, and state set, **Ci**.

Therefore, Sequence (1) is described as follows:(*c*10, *c*20), (*f*1(*c*10, *c*20), *f*2(*c*10, *c*20)), …
Defining (*f*1(*x*, *y*), *f*2(*x*, *y*)) ≡ (*f*1, *f*2)(*x*, *y*) ≡ *F*(*x*, *y*), Sequence (1) is represented as follows
(*c*10, *c*20), *F*(*c*10, *c*20), *F*(*F*(*c*10, *c*20)), …, *F^n^*(*c*10, *c*20), …,(2)
which can be written as follows: *u*0, *F*(*u*0), *F*(*F*(*u*0)), …, *F^n^*(*u*0), …,
where *u*0 = (*c*10, *c*20), *F*: **U** ➝ **U**, **U** ⊂ **C1** × **C2**; in general, **U** ⊂ ∏ **Ci**, where “*i*” denotes component cognizers. *F* is the cognition function of the world *U* with state set **U**.

Cognizers do not exist as entities in physical space like containers; instead, they themselves are state spaces (**Ci**) constituted by their cognitive functions, manifesting their properties that relate each state with others within and between state sets (Figure 4A).

The world is isolated because it has nothing outside to cognize, and whose state changes are determined through internal cognitions among its components. It may be controversial to call *F* a cognition function since it has nothing to cognize; however, we refer to it as a special form of cognition function because *F* is constituted by cognition functions of components, such as *f*1, *f*2, and so on. 

The CS model is a general framework that may be useful for addressing interdisciplinary issues, such as those discussed in this study. However, it remains unclear how the CS model can be translated into specific models in physics, chemistry, and biology; specifically, how **Ci** and *fi* can be translated into other models (languages). For example, the relationship between a quantum system and its measurement is an important issue involving various interpretations that are still debated. Cognition is defined as a related state change in the CS model, encompassing both the state determination occurring in a system and that occurring in measurement. In this study, how this “cognition” in the CS model corresponds to the collapse of quantum superposition has not been elucidated. This should be investigated in future research.

### 6.3. Selectivity and Discriminability of Cognition

The cognition function for each cognizer involves two properties concerning how it changes its current state in relation to other states in the world: selectivity and discriminability (Section 4). The determination of a succeeding state, represented by a cognition function, implies the selection of one succeeding state (e.g., *c*1*j*), which narrows down the relation to the other cognizer, represented by a subset of the direct product **C1** × **C2**, specifically “(*c*1*j*, ?)” (Figure 4C). The full determination is completed by all component cognizers.

Any cognizer cognizes all other cognizers wherever they exist. Cognition includes both discrimination and non-discrimination without violating the causal principle (Section 6.4). Discrimination refers to a change from a given state to different states in response to different environmental states. Suppose that two or more different environmental states occur, for example, *c*2*i*, *c*2*i*′, when a focal cognizer is in a given state, *c*1*i*. When *f*1(*c*1*i*, *c*2*i*) ≠ *f*1(*c*1*i*, *c*2*i*′), *C*1 in *c*1*i* discriminates between the environmental states *c*2*i* and *c*2*i*′. 

Discriminability affects the uncertainty in events occurring to the cognizer: if *C*1 does not discriminate between different environmental states, *c*2*i* and *c*2*i*′; i.e., *f*1(*c*1*i*, *c*2*i*) = *f*1(*c*1*i*, *c*2*i*′), *C*1 will face uncertainty about the environment, *f*2(*c*1*i*, *c*2*i*) or *f*2(*c*1*i*, *c*2*i*′), as shown in Figure 4D. If *C*1 discriminates between them by changing different states in response to *C*2’s different states, *C*1 will experience a unique *C*2 state following the cognition (*c*1*i* ⟼ *c*1*j*). Discrimination reduces the uncertainty of events. However, LSs should ignore unimportant differences and process only those differences relevant to maintaining favorable external and internal relationships.

“Discrimination” focuses on differences in state, not on the presence or absence of a target entity within the system. However, this presence or absence can be represented as a discrimination between different positions (states) of a target entity, located within and outside the locality (see Section 6.7 in relation to local causation).

### 6.4. Causal Principle (The Principle of Causality) and Freedom

Causality refers to the rule-based determination of a succeeding state from a previous state of the world. The “causal principle” (or, the “principle of causality”) postulates that, given a state of the world, the succeeding state is uniquely determined by the property of the world from a current state (do not confuse “causality” with “causation”). This is called causal determinism. No external agent determines changes in the world state. In other words, the CS model assumes that cognizers determine state changes of the world. Their behavior is not free from the states of others; instead, each cognizer participates in world dynamics by determining (selecting) their succeeding states based on their properties.

The causal principle is a metaphysical assumption because this principle cannot be falsified empirically [10]. Scientific theories may or may not adhere to this principle. The principle is formalized as follows. Given two states of the world, *ui* and *ui*′ (*ui*, *ui*′ ∈ **U**), if *ui* = *ui*′, then *F*(*ui*) = *F*(*ui*′). That is, given a state of *U*, only one succeeding state is determined. 

This principle can be expressed in terms of the CS model. Consider a focal cognizer (*C*1) and its environment (*E*), with cognition functions *f*1 and *fe*, respectively. The environment is a cognizer, which may be a complex of cognizers external to the *C*1. If (*ci*, *ei*) = (*ci*′, *ei*′), where *ui* = (*ci*, *ei*) and *ui*′ *=* (*ci*′, *ei*′), then (*f*1(*ci*, *ei*), *fe*(*ci*, *ei*)) = (*f*1(*ci*′, *ei*′), *fe*(*ci*′, *ei*′)), meaning that the successor states of *C*1 and *E* are, respectively, the same. Indeterminism allows *f*1(*ci*, *ei*) ≠ *f*1(*ci*′, *ei*′) when (*ci*, *ei*) = (*ci*′, *ei*′), meaning that different *C*1 states can follow the same states of *C*1 and *E* (the same applies to *E*). In the above representation, *F*: **U** ➝ **U**; *f*1: **C1** × **E** ➝ **C1**; *fe*: **C1** × **E** ➝ **E**.

### 6.5. State and Event

State and event are different concepts. The meta-observer defines the states in such a way that each state can be differentiated from the others. Events occur to a cognizer through cognitions of the environment as related state changes (*ci* ⟼ *cj*). Therefore, events are subject-dependent, meaning that a focal cognizer experiences events through cognition. When a molecule is a focal cognizer, it cognizes (i.e., changes its state in relation to) other cognizers, such as subatomic, atomic, and molecular entities. In this context, a conformational state change is an event that occurs for this molecule through its cognition (i.e., chemical cognition) of the environment. When an experimenter (as a focal cognizer) observes this molecular change, an event occurs to him through his cognition (i.e., a state change in relation to the molecular process). 

A cognizer may cognize two or more states of the environment as the same. In such cases, the same event can occur for the cognizer in various environmental states. For instance, “heads-up” in coin tosses is the same event for an observer, even though “heads-up” includes different positional states of the coin when it lands on the ground. Another observer may cognize “heads-up” as different events, depending on the landing location.

### 6.6. Causation (Cause–Effect Relationship)

Causation refers to a specific type of relationship in state change between two entities. In the CS model, the cause-and-effect relationship, or “causation”, refers to discriminative state changes (“effects”) in response to different external states (“causes”). Formally, this is represented as *ci* ⟼ *cj* or *cj*′ in response to *ei* or *ei*′, respectively (where *cj* ≠ *cj*′ and *ei* ≠ *ei*′). Non-discrimination is considered non-causal or “ignoring”. For instance, if *ci* ⟼ *cj* occurs in response to *ei* or *ei*′, it indicates that the cognizer does not discriminate between the states.

If a cognizer (A) does not discriminate between different states of a nonlocal cognizer (B), it means that A cognizes B but is not causally affected by B’s state. Non-discriminative cognition is similar to the concept of “zero”, which, while representing nothing (i.e., the absence of quantity) in some contexts, has unique properties as a fundamental element of the number system in algebra. 

For example, a rolling ball changes its velocity upon colliding with another ball. In this case, the first ball discriminates (as an effect: *cj* or *cj*′) between different states of the second ball (as a cause: *ei* or *ei*′), indicating the second ball causes the velocity change of the first ball. Similarly, a receptor protein changes its molecular conformation when it binds to a ligand. Here, the protein discriminates (as effects) between different states (as causes) of the ligand, indicating that ligand binding causes the receptor’s conformational change as an effect. 

### 6.7. Principle of Local Causation

The principle of local causation asserts that causal effects can occur only within a locality, denying action at a distance. This is a standard assumption in the physical world (though it does not deny non-local correlation), and while optional, it is included in the CS model. Locality, as a relationship between *different* cognizers, can be defined using the concept of discriminability. According to the principle of local causation, cognizers do not discriminate between different states of cognizers that are distant (non-local) in position. This is based on “causation as a discriminatory relationship”. State correlations are generated through discriminative cognitions by component cognizers. As described previously (Section 4), under the principle of local causation, semiotic cognition plays a vital role in discriminating between the states of distant cognizers, mediated through state correlations among cognizers.

### 6.8. Hierarchical Structures of Cognizers Systems

LSs form a nested hierarchical organization [84,85]. There is no consensus on how the living world is hierarchically organized, which varies according to the researcher’s purpose. The nested hierarchical structure is based on part-whole relations. Nakajima [17] clarified the distinction between two types of part-whole relations: synchronic and diachronic, which have often been confused and not distinctly recognized in the literature.

The synchronic part-whole relation is a standard one, referring to a relationship in which the whole is composed of parts synchronically; parts continue to exist without appearing or disappearing over time. The representation of a cognizers system comprising *C*1 and *C*2 (or *E*) is synchronic. However, in LSs, such as single-cell or multi-cell organisms, constituent molecules are renewed through catabolism and anabolism, exchanging atoms and molecules across the membrane with the surrounding medium. This self-making mechanism has been described using various terminologies by different authors [19,20,21,22,23]. The “processes maintaining a particular organizational pattern” by renewing constituent lower-level components can endure for certain periods and can be recognized as a “diachronic whole” generated by transient entities as its parts. Synchronic cells are similar to frozen cells, whereas living cells are diachronic cells, each retaining a specific identity in the process generated by the generation and degradation of parts. Such a process can be described as an entity (cognizer) with a new state space, that is, the density distribution of atoms and molecules. In other words, a living cell can be described as a cognizer that is synchronically composed of sub-cognizers, existing in their density spaces (see Section 8.1). This description is similar to field concepts in physics [17].

Unlike Leibniz’s Monads, the CS model provides no fundamental cognizers as the most basic units (atoms) of the world. Monads are the basic units of the metaphysical universe. Cognizers can be defined relative to other cognizers in a hierarchical manner. Additionally, unlike cognizers, Monads are causally closed to external entities: ‘the natural changes of the Monads come from an internal principle since an external cause can have no influence upon their inner being’ [86] (Section 11).

## 7. Internalist Model of Semiotic Cognition toward Unification of Matter and Mind

### 7.1. The Problem of Escaping Solipsism for LSs

Under the principle of local causation, physical and chemical entities (cognizers) are blind to non-local external states. However, LSs must respond to external states, local or non-local, appropriately. As discussed in Section 4, LSs can discriminate between different states of distant objects by semiotic cognition (here, “states” means its relative states to the subject, not absolute). For example, the photoreceptor cells of an animal discriminate between particular states of the electromagnetic field by receiving photons in the locality, and the organism can discriminate between states of a distant object relevant to survival/reproduction based on a state correlation between photons and the object. To achieve these tasks, LSs must (i) measure the local external states, (ii) produce symbols that signify the local external states, (iii) produce symbols, based on the symbols for local states, that signify the distant external states that are relevant to survival and reproduction, and (iv) produce effector state changes to relate themselves to the surroundings. 

From an external observer’s perspective, discriminative state changes (*xi* ⟼ *xj* or *xj*′) of a given cognizer measure the external states, *ei* and *ej*, under the condition that they occur from the same state (*xi*). However, from the internal perspective, it is unclear what function guarantees that the sensor’s state changes (*xi* ⟼ *?*) indicate the external states. This function requires a special kind of system that operates on the inverse causality principle, and a device for measuring the relative states of the environment, called the “measurer”, which is explicated in the next section.

### 7.2. Semiotic Cognition by Inverse Causality Operation

Measurement by inverse causality can solve this problem [12,15,16]. “Inverse causality” is an epistemic principle operating in a subject system to produce internal states as symbols that signify past states of external reality hidden to the subject (note that epistemic and ontic state changes are unified as cognitions in the CS model). Inverse causality is not “reverse causation”, “retrocausality”, or “backward causation”, asserting that future (or present) states affect past states or that their effects produce causes.

To understand inverse causality, we begin with the unique successor principle, which postulates that any element in a sequence has only one successor. Formally, if *xi* = *xi*′, then *G*(*xi*) = *G*(*xi*′), where *G*(*xi*) and *G*(*xi*′) are the successor elements of *xi* and *xi*′, respectively. This principle is applied to any sequence of elements but is identical to the causal principle (Section 6) when it is the sequence of a temporal state change in a system. The contraposition of this principle in temporal state sequences is called “inverse causality”, which postulates that if *G*(*xi*) ≠ *G*(*xi*′), then *xi* ≠ *xi*′, meaning that any two different states have different predecessors (*xi* and *xi*′). 

Husserl considered phenomena as mental processes and analyzed the stream of these processes to clarify the relationship between the mind and the world [87,88]. From the internalist framework (see Section 5: Experiment *E* for an illustration), the environment for a given LS is neither assumed nor denied, similar to how Husserl suspended judgment of, i.e., bracketed, the existence of external things in his phenomenology. 

Consider sensor *A*, a component of LS, which is in state *a*0. From an external observer’s perspective, the sensor’s state changes from *a*0 to *ai* in response to the external state, *ei*, and from *a*0 to *aj* in response to the external state, *ej* (Figure 5A). In this case, the sensor discriminates between *ei* and *ej* when in *a*0. This process is described from the external perspective. From the internal perspective, the external states, *ei* and *ej*, are hidden, and the subject LS have only the following processes: *A* changes from *a*0 to *ai* in some instances and from *a*0 to *aj* in others. 

According to inverse causality, if *ai* ≠ *aj*, their predecessors must differ. However, in either case, the sensor state is *a*0, which violates inverse causality. The occurrence of *a*0 in either case is *certain* for the subject system and cannot be altered. Therefore, by introducing different symbols *bi* and *bj* behind *a*0 (Figure 5B), the entire state becomes different, i.e., (*a*0, *bi*) ≠ (*a*0, *bj*). The state changes now fulfill inverse causality for that part and satisfy the causal principle when replayed forward in time: (*a*0, *bi*) ⟼ *ai* and (*a*0, *bj*) ⟼ *aj*. *bi* and *bj* are the states produced internally as the states of another component (cognizer), say *B*, of the LS. By comparing Figure 5A with Figure 5B, we can understand that *bi* and *bj* are symbols (internal states) that signify “*ei*” and “*ej*”, respectively. This operation is called “inverse causality operation (IC operation)”, or simply “inverse causality” when it is not confusing. Note that *bi* and *bj* are states of *B* that must occur from the same state (*b*0) to perform the IC operation to measure the states of *A*. 

The IC operation is the measurement by inverse causality, an internal process (black arrows in Figure 5) by which a system discovers (realizes; makes it real) external states hidden to the system. A replay of this IC operation forward in time indicates a causal process between the subject system and its environment from an external perspective. Here is a chicken-and-egg problem: do the data confirm the external reality, or does the external reality cause the data? [2] (Chapter 6). The logical equivalence between inverse causality and causality may answer this problem: they are the same, but the reality is subject dependent, which is called the “equivalence principle in the subject-reality relationship” [16]. Under this equivalence, the new elements derived from the IC operation signify the external states for a subject system. 

Any device used to measure external states by IC operation must have the property of returning to the same baseline state (such as *a*0, *b*0, and *c*0 in Figure 5) after the state changes. Such a device is called a “measurer”. Measurer is a particular class of cognizers that perform physical or chemical cognitions and return to their baseline state. LSs invented many kinds of measurers, including molecular measurers such as cell-surface receptors, ion channels, transporters, molecular switches in signal transduction within and between cells, and neural cells in multicellular organisms [89,90]. They may have more than two baseline states; however, the IC measurements can operate correctly when they change from the same baseline [16].

The change from a baseline state (*a*0) to *ai or aj* is shown in Figure 5B. Similarly, assume that *B* is also a measurer with a baseline state (*b*0). This indicates that *B* changes from *b*0 to *bi* or *bj*. These changes violate inverse causality. Therefore, different symbols are introduced behind *b*0, such as c*i* and c*j*, to fulfill inverse causality, as shown in Figure 5C. LSs are composed of physical and chemical entities (cognizers) that are subject to the principle of local causality (Section 6). Therefore, local external states can only be *directly* measured by them. For example, a fly’s sensor measures the local state of molecules (a sign of food) released from a distant food. However, LSs can measure nonlocal states *indirectly*, i.e., mediated through direct measurements (*bi* and *bj*), as illustrated in Figure 2D. Here, consider another measurer, *C*, which takes in states that signify the external states (e.g., *ek* and *el*, i.e., the states of food) that cause *B*’ state changes, *b*0 ⟼ *bi or bj* by IC operation (*A* cannot detect the difference between *ek* and *el*) (Figure 5C). *C*’s states (*ci* and *cj*) signify states of distant entities (food) that *A* cannot detect, that is, *a*0 ⟼ *a*0 when *b*0 ⟼ *bi or bj* (not represented in Figure 5C), which can be discriminated by an external observer. (Note that *bi* is not a state of the molecule but a B’s state as a symbol that signifies a state of the molecule. The same thing is true for *ci*). The aforementioned formalization of the IC operation focuses on knowing the external states. However, the same operation can also be used to measure and process the internal states of the non-IC components of an LS, such as interoception for homeostatic regulation. 

Scientists use meters for observation [91]. Experimenters use a meter (e.g., an ammeter) to measure the physical state of an object that cannot be measured by the senses. In this case, they first produce a mental state (e.g., “15 mA”, corresponding to a *B*’s state) that signifies the states of the meter (a sign as a mental state), and then produce a secondary mental state (*C*’s state) that signifies a hidden physical state (a mental state signifying a state of an external object: electric field). The use of meters may be a major reason why physical entities usually exhibit *quantitative* states. 

The IC operation in the internalist framework is similar to a specific type of information introduced by physicist John Wheeler. He declared that “every it—every particle, every field of force, even the space-time continuum itself—derives its function, its meaning, its very existence entirely—even if in some contexts indirectly—to yes or no questions, binary choices, bits” [92]. He called the process of finding reality “it from bit”. In contrast, the orthodox framework of information science takes an externalist framework (i.e., an external observer’s perspective), similar to Shannon’s information [93]. A subject entity receives information (a message) quantified in a bit from an information source, which might be dubbed “bit from it”. Here, “it” and “bit” (datum) a receiver gets are observed by an external observer. 

### 7.3. The Measurement System by IC Operation

The CS model has an externalist framework because it describes a focal cognizer and its environment, which includes other cognizers in the entire system. From the internalist perspective, only the subject system exists. However, the CS model can be easily modified into the internalist model by focusing on a single cognizer, such as a multicellular organism or a single cell, as “a system of sub-cognizers” without explicitly assuming the environment. In the following, we focus on a single cognizer system (no “s” for cognizer) that incorporates sub-cognizers performing the IC operation to produce symbols that signify external states, which are processed for actions.

The IC operations illustrated in Figure 5B,C are logical, algorithmic processes that proceed backward in time (see arrows in the figures). Therefore, to apply the logical process to an LS’s semiotic cognition, we must transform the process into a cognizer (epistemic and ontic) process moving forward in time [16]. Figure 6 illustrates how semiotic cognition in the internalist framework can operate by a system of measurers, performing physical or chemical cognitions, producing symbols that signify the external states, and yielding final actions. The system requires a specific cascade (i.e., non-nested hierarchical) arrangement of measurers that can perform IC operations, as shown in Figure 6A. 

We consider our classes of measurers, including *A^M^*, *B^M^*, *C^M^* (each corresponding to *A*, *B*, and *C* in the previous model in Figure 5), and *D^M^*, which is added here.

*A^M^* is a sensor whose states are data. Data are given because they are not produced (entailed) by something else in the internalist framework (no external reality is assumed).*B^M^* measures *A^M^* states. The *B^M^* is the reader of the data (*A^M^* states). *B^M^* states are symbols that signify local external states. Examples of *B^M^* may be second messengers of signal transduction in cells.*C^M^* measures *B^M^* states, whose states signify non-local external states that cannot be derived by inverse causality in *A^M^*-*B^M^* coupling. *C^M^* states are not copies of the *B^M^* states. Therefore, *C^M^* interprets (decodes) the *B^M^* states. *C^M^* links the *B^M^* to the *D^M^* (see below) semiotically, leading to a final action. In some cases, *B^M^* states may be linked directly to *D^M^* without mediation by *C^M^* (see Section 8.3 for examples).*D^M^* is the effector, a measurer that transforms the symbols produced in *B^M^* or *C^M^* to structural changes of the subject LS in a particular manner. For example, the effector may include the translation of enzyme proteins in gene expression systems (Appendix C) and locomotor apparatus, including motor neurons and muscle cells.

These measurers do not necessarily involve single molecules or neurons. Instead, they are collections of molecules in the density space of a single cell or a collection of neural cells (see Section 6.8). In other words, *A^M^*, *B^M^*, *C^M^*, and *D^M^* can be composed of sub-measurers (Figure 6B). A combination of sub-measurers’ states can represent the symbols of external states. The upstream symbols are integrated and differentiated, and then mapped into subsets of downstream measurement symbols. Such structures are embedded in the signal transduction systems within and between cells, and also in neural systems.

*A^M^* changed from baseline *a*0 to *a*1 at *t*2. From this change, the measurement system produced “*b*1” as the *B^M^* state through an IC operation. Here, *b*1 signifies an external state (“?_b1_” in Figure 6A) hidden to the subject at *t*1. Then, *C^M^* discriminates *B^M^*’s states, changing to *c*1, where *c*1 signifies an external state (“?_c1_”) hidden to the subject at *t*2 (see also Figure 5C). *D^M^* does not signify the external states; instead, it transforms *C^M^* states into the effector (*D^M^*) states. The state changes in each measurer are linked in a specific relationship in the *A^M^*-*B^M^*-*C^M^*-*D^M^* coupling. For any measurer, other measurers are internal to the system. Each measurer, excluding *A^M^*, discriminates between the states of the preceding measurer in the cascade. *A^M^* alone has nothing to discriminate between differences *within* the system. In other words, the inner process does not entail changes in the state of *A^M^*. Through IC operation, the system produces symbols that signify the external states that cause *A^M^* state changes when viewed from an external observer.

Note that in Figure 6A, there may be *a*0 ⟼ *a*2, *a*0 ⟼ *a3*, …, downstream in the diagram (not represented in the figure), for which various states of *B^M^* and *C^M^* can be produced, and various *D^M^* states (effector actions) may follow. How measurers produce symbols in the form of their states determines the possible events occurring in *A^M^* (e.g., *ax* in Figure 6A), which concerns adaptations to its surroundings.

In the externalist framework, the structures of the IC system, as adaptations, are embedded in signal transduction systems in cells or neural systems in animals. Coupling structures between measurers can develop in an LS within a generation through learning processes, in which the mapping of upstream symbols into downstream symbols is modified. Furthermore, over generations, if there is a variation in the IC measurement systems in a population, some coupling structures will become dominant or diversified as adaptive radiation through natural selection. 

In the IC measurement model, “knowing the external world (environment)” is defined as the process of a subsystem of an LS in which (i) the internal states are produced as symbols that signify external states (i.e., sensation), (ii) they are processed to produce states of the intermediate component (i.e., perception or interpretation), and (iii) they lead to changes in the state of the final component as effects (actions including gene expression, metabolism, and behavior). These three processes are performed by IC measurements via *A^M^*-*B^M^*, *B^M^*-*C^M^*, and *C^M^*-*D^M^* couplings, respectively (Figure 6A,B). The entire process involving symbol production and subsequent state changes is conceptualized as “semiotic cognition”.

Multicellular organisms have evolved neural systems for IC operations to detect hidden external states, as well as internal states (i.e., interoception). Each neural cell functions as a sub-measurer, with a resting potential as a baseline state, which can be depolarized to an activated state. When measurers *A^M^*, *B^M^*, and *C^M^* are composed of many neurons, as shown in Figure 6B, there is a variety in the abundance of sub-measurers and their connections with sub-measurers in contiguous layers.

### 7.4. Adaptation and the Production of Symbols That Signify External Objects

The philosophy of perception investigates the nature of our sensory experiences and their relation to reality [94]. Organisms must identify their environmental states to which they orient themselves and act to survive longer and reproduce more. This problem of perception is relevant to the focus of this study, as it highlights the connection between LSs’ semiotic cognition and their adaptation to their environments. 

In the field of cognitive science and philosophy of mind, there is an ongoing discussion regarding whether perception estimates the truth about external reality. Given a person or an organism in general, a conventional idea asserts that organisms whose perceptions are more accurate and are fitter to survive and reproduce than those with less accurate perceptions. In other words, natural selection favors “veridical perceptions”, those that more accurately represent the external states [95,96,97]. 

In contrast to the prevailing view, Hoffmann proposed the interface (desktop) theory of perception, which asserts that ‘our perceptions constitute a species-specific user interface that guides behavior in a niche’ [98,99]. Just as the icons of a PC’s interface hide the complexity of the computer, our perceptions usefully hide the complexity of the world and guide adaptive behavior. Mark, et al. carried out a computer simulation to investigate this hypothesis, concluding that ‘veridical perceptions can be driven to extinction by non-veridical strategies that are tuned to utility rather than objective reality’ [100]. This suggests that natural selection does not favor veridical perceptions. ‘Whereas naive and critical realism assert that perception is useful because exhaustively or in part, it is true, the desktop theory asserts that perception can be useful because it is not true’ [100]. 

However, previous endeavors have not been successful in addressing the problem of metaphysical solipsism, which refers to the challenge of verifying the existence of “external reality” to which LSs must adapt, without mistaking perceptions or icons that appear in hallucinations or dreams for those that pertain to external objects. Inverse causality has been proposed as a potential solution to address this challenge. The IC model posits that external reality exists if it has various states over time, which can be detected (entailed) through the IC operation of given data. (Note that IC systems can also operate in response to internal states, that is, inside the body. Here, the processes are triggered by internal states to maintain an appropriate internal order, known as interoception).

As illustrated in Figure 6A, an IC system using *A^M^*-*B^M^* coupling detects the external reality in distinct states, conceptually corresponding to “sensation”. *B^M^*-*C^M^* coupling corresponds to “perception” in a conventional sense. The equivalence principle (Section 7.2) allows symbols derived from the IC operation to represent the external states that cause the data. This equivalence bridges between internalist and externalist perspectives. However, if some of these measurers malfunction due to pathological issues or mutations, they cannot correctly produce symbols that signify different external states. For example, if the measurers do not return to the same baseline state, differences cannot be derived. Measurers must have a non-nesting hierarchy, such as *A^M^*-*B^M^*-(*C^M^*)-*D^M^* coupling, to perform “semiotic cognition”. If an organism loses this structure, it will result in the malfunction of semiotic cognition. For example, if an organism has an abnormal IC system characterized as X-*B^M^*-(*C^M^*)-*D^M^*, in which “X” refer to non-measurer components of the organism, it cannot correctly sense and interpret external states, leading to “hallucinations” (i.e., sensations that are not caused by external states). 

The external reality in the above description is a set of states of the entire environment that is not differentiated into component objects in the environment. This entire reality external to an organism measured by IC operation is called “proto-reality” [12,16]. However, as illustrated in Figure 6B, each measurer (*A^M^*, *B^M^*, *C^M^*, and *D^M^*) can be differentiated into sub-measurers, which can create a variety of sensations, percepts, and behavioral actions. Suppose that *A^M^*, *B^M^*, *C^M^*, and *D^M^* are composed of sub-measurers: *A*_1_*^M^*, *A*_2_*^M^*, *A_3_^M^*, …, with state sets **A_1_^M^**, **A_2_^M^**, **A_3_^M^**, …; *B*_1_*^M^*, *B_2_^M^*, *B*_3_*^M^*, …, with state sets **B^M^_1_**, **B^M^_2_**, **B^M^_3_**, …; *C*_1_*^M^*, *C*_2_*^M^*, *C_3_^M^*, …, with state sets **C_1_^M^**, **C_2_^M^**, **C_3_^M^**, …; and *D*_1_*^M^*, *D*_2_*^M^*, *D_3_^M^*, …, with state sets **D_1_^M^**, **D_2_^M^**, **D_3_^M^**, …, respectively. For example, the direct product of **B_1_^M^ × B_2_^M^** may signify the state set of a particular object, that is, a component of the proto-reality. Similarly, the direct product of **C_2_^M^ × C_5_^M^**, for example, may signify the state set of a particular nonlocal object. The manner in which these sub-measurers are combined indicates how the organism divides the entire set of symbols signifying the states of proto-reality into those of individual objects. 

The differentiation of the proto-reality, an entire environmental whole, into individual object entities is closely related to Saussure’s linguistics (semiotics) [101]. The concepts signified in a language are arbitrary divisions of a continuum, which are not autonomous entities; they are divided in ways particular to the language it belongs to [102]. Thus, the internal construction of external objects is similar to the concepts of “sign” by Saussure, “umwelt” by Uexküll, and “icon” by Hoffman. In the box-play game from the internalist perspective (see Thought experiment E in Section 5.2, Figure 2E), balls of various colors are conceptual objects created internally by a player. Some players may treat orange and red balls as same-colored balls to which they act.

### 7.5. Where Is the Mind in a Cognizer?

Leibniz doubted that feeling and thinking can be explained by mechanism, suggesting that there are basic atomic components (such as Monads) that must have the ability to perceive. In his book Monadology [86] (Section 17), he claimed: ‘Moreover, it must be confessed that perception and that which depends upon it are inexplicable on mechanical grounds, that is to say, by means of figures and motions. And supposing there were a machine, so constructed as to think, feel, and have perception, it might be conceived as increased in size, while keeping the same proportions, so that one might go into it as into a mill. That being so, we should, on examining its interior, find only parts which work one upon another, and never anything by which to explain a perception. Thus, it is in a simple substance, and not in a compound or in a machine, that perception must be sought for’. 

In the above IC measurement model, we can find only physical or chemical cognition by the measurers, just as we observe only physical and chemical processes in the brain. Where is consciousness or awareness? This is known as the hard problem of consciousness [103]. Assuming a mental property at the primary (physical or chemical) level cannot solve our problem, because how semiotic cognition, as a mental process that involves feeling and thinking processes, can arise from the primary level remains unclear. As explained in the IC operation based on the CS model (Figure 6A,B), semiotic cognition can occur from a system of measurers (a special type of cognizer, as defined above) organized with a specific cascade (hierarchical) structure in the measurer coupling, as illustrated. These systems can produce internal states as symbols that signify external states and are processed for actions. This property satisfies the necessary condition for consciousness, that is, being aware of the surroundings, which is part of the adaptive function of LSs that is not restricted to multicellular organisms with neural systems but includes unicellular organisms (see Section 8).

Consciousness is not a construct that is more certain than external reality, as suggested by Descartes’s skepticism [12,104]. According to this skepticism, the brain might be a model constructed in the mind. The external world (environment) could be an illusion. To solve the chicken-and-egg problem of mind-reality circularity, the hierarchical extension of cognition may unify the mind and material entities in the monistic framework of the CS model, avoiding a mind-only or matter-only world. 

The definition of consciousness varies among researchers. From a biological point of view, consciousness should be defined as having evolved from non-conscious beings due to adaptive functions, not as a byproduct of biological evolution. Therefore, in this study, we posit that being aware of the surroundings and the inside (i.e., knowing external and internal states and responding accordingly) is the core function of consciousness and the necessary condition for something to have consciousness. Perceptual contents, including qualia of mental states, such as “there is something it is like to be” [105], arise from the performance of this core function. The core function of consciousness is almost identical to discovering a way to escape solipsism. 

Let us introduce a working definition of consciousness as awareness, based on the IC model. Hypothetically, semiotic cognition through IC operation represents the mental process, with some parts being conscious. Accordingly, humans may understand other people’s subjective experiences in terms of their own experiences. This is because they have similar isomorphic IC systems. For example, for people using the same traffic signal system, “*b*1” and “*c*1” in Figure 6A may have the same qualia, such as “red” and “dangerous if not stop”, respectively, leading to braking action (“*d*1”), although the redness and dangerousness in their perceptions might not be identical but have the same role in forming their relationships with the hidden surroundings (others’ cars and people).

One cannot think that a billiard ball, a cognizer at the physical level, sees the surroundings, feels pain when colliding with a wall, and flees, because the ball does not have an IC operation system (i.e., it is non-conscious). What about bacteria? Are bacteria conscious of (or equivalently aware of) their surroundings? We cannot answer this question without first defining consciousness. Additionally, an explicit model is required for testing, in which consciousness is defined operationally. In the CS model with IC operation, “consciousness” is replaced with a scientifically tractable concept, that is, the production of internal states as symbols signifying the external states, being processed for actions (Section 1, Appendix A). In this sense of “consciousness”, bacteria may have it because they perform the IC operation in chemotaxis and quorum sensing. However, their consciousness is far from human consciousness, because the structure and complexity of their IC measurement systems are not isomorphic. IC systems in unicellular organisms are composed of a small number of sub-measurers (molecules) in each layer (Figure 6B); thus, their interconnections appear to be quite simple (see Section 8.3 for details).

Furthermore, one may ask whether an ecosystem is conscious (in the sense of IC operation) at a higher level. However, this is not so because no ecosystem has an IC system at this level. Similarly, one may ask whether a group of organisms is conscious. Suppose a human social group, in which some persons act as sensors and some as messengers who bring sensor data to others as interpreters who determine how group members should behave. Group-level consciousness may exist in terms of IC operations. However, this IC system is much simpler than that of the individuals (their brains) in this group. If consciousness is defined as a human-equivalent IC system that can manage complex languages, then only humans are conscious. 

Cognizers’ states, such as *xi*, *c*1*i*, *a*0, *b*1, …, used in the CS model, are all symbols in the meta-observer’s languages, from which their qualia, if any, have been removed by abstraction. Humans, animals, plants, microbes, molecules, and subatomic particles/fields are cognizers in the CS model. Their states might have certain kinds of qualia, but these cannot be experienced directly by others by definition. This proposition is not testable and can only be assumed by a model.

## 8. IC Measurement Systems in Unicellular LSs

In the previous section, we explained how LSs perform semiotic cognition using IC measurement systems, without recourse to external perspectives. Next, we address a possible scenario in which chemical autopoietic systems evolutionarily invented IC measurement systems for semiotic cognition, beyond the limitations of physical and chemical cognitions, to manage the probability distribution of events as adaptations.

### 8.1. Molecular System as Cognizers System

Before describing the scenario, we outline how chemical systems can be translated into the framework of cognizers systems. Consider molecules that behave in state space in terms of their positions and velocities (Figure 7A). Each molecule, as a “particle”, can be described as a cognizer because it has a property according to which it determines a succeeding state depending on its current state and the surrounding state. These molecules can be classified into subpopulations: type *a* (*a*_1_, *a*_2_, …), type *b* (*b*_1_, *b*_2_, …), and type *c* (*c*_1_, *c*_2_, …). Here, individual molecules of the same type (e.g., *a*_1_, *a*_2_, …) *synchronically* constitute a subpopulation of molecules {*a*_1_, *a*_2_, …}. This is a *synchronic* part-whole relationship between cognizers at the focal and contiguous higher levels. 

However, individual molecules (particles) can be generated and degraded over time through chemical reactions. As stated in Section 6.8, in this situation, a subpopulation of molecules of the same type (e.g., *a*_1_, *a*_2_, …) can be described as a single “field” entity (e.g., *a*) at a higher level that behaves in a density-state space over a positional space, as shown in Figure 7B. Each molecule, as a field, can also be described as a cognizer that has its own state space (spatial distribution of density) and the property that determines state changes. Individual molecules as “particles” of the same type (e.g., *a*_1_, *a*_2_, *a_3_*, … in Figure 7A) *diachronically* comprise a molecule as a “field” (e.g., *a* in Figure 7B) through generation and degradation over time; this is a *diachronic* part-whole relationship between cognizers at the focal and contiguous higher levels. Furthermore, molecules (fields) such as *a*, *b*, and *c synchronically* constitute an entire molecular field.

As shown in Figure 7C, metabolic closure may emerge as a pattern formed by molecules as fields (*a*, *b*, *c*, *d*, and *e*) in a density-state space under particular physical and chemical conditions (Figure 7C). Closure can be recognized as an autopoietic system with a dynamic border that is synchronically constructed by molecules as fields (it is a diachronic construct of molecules as particles). If a metabolic closure is separated from others by a membrane made of molecules produced by metabolism (e.g., “*g*” in Figure 7C), it can allow some molecules to selectively enter (or let them go outside) by physical or chemical cognition determined by the properties of molecules composing the membrane. Additionally, the membrane can physically prevent diffusion of the component molecules of metabolism to maintain spatial localization and interactions between component molecules.

### 8.2. Chemical Systems Operating with Chemical Cognition

Protocells are metabolic and self-replicating systems that operate by physical and chemical cognitions through prebiotic, generating protocells via chemical evolution in the geochemical environment of ancient Earth [58,83,106]. Quantum measurements in the molecular world must have been operating in synthetic chemical reactions, leading to the origins of life in a prebiotic environment [107,108,109]. 

Metabolic closure is a fundamental property that categorizes systems as living (Section 3). Autopoietic chemical systems produce nearly all system components (molecules) through a chemical network of composition and decomposition through chemical cognition. Metabolic networks may be facilitated by a catalytic process that generates specific chemical reactions at relatively low molecular concentrations at relatively low temperatures and pressures [110] (Chapter 5).

There is controversy regarding whether autopoietic metabolism is cognitive. For example, Bitbol and Luisi [111] claimed that metabolism always involves dynamic interactions with the outer medium, suggesting that ‘a full-blown metabolism is tantamount to cognition’. However, as Bourgine and Stewart [112] argued, being autopoietic can be distinguished from being cognitive. According to them, autopoiesis focuses on the internal function of metabolism, which is not cognitive (although it involves the regulation of the boundary conditions necessary for its ongoing existence as a metabolic network), whereas cognition focuses on ‘management of the interactions between an organism and its environment’. This controversy can be resolved by distinguishing between cognition at the chemical and semiotic levels. Autopoietic systems are cognitive at the level of molecular systems that operate with chemical cognition, which are not semiotic. 

### 8.3. How Semiotic Cognition Can Emerge in Chemical Systems

Bourgine and Stewart [112] divided the interactions between a system and its environment into interactions that have consequences for the internal state of the organism (type A interactions, i.e., “sensations”) and interactions that have consequences for the state of the local environment or that modify the system’s relation to its environment (type B interactions, i.e., “actions”). They then defined a system as cognitive ‘if and only if type A interactions serve to trigger type B interactions in a specific way, so as to satisfy a viability constraint.’ However, this model describes ‘interactions between a system and its environment’ from the perspective of an external observer, which is not from a focal system. It remains unclear how a focal chemical system can ensure that a given internal state change (e.g., that occurring in a sensor) is caused by interactions with the environment.

How can type A interactions lead to differences in external states? The IC measurement model (Section 7.3) solves this problem. Notably, IC measurement is based on the same baseline state from which a measurer changes its state, although measurers can possess multiple baseline states. To illustrate this, consider a malfunctioning measurer that cannot maintain the same baseline state as the reference state. If the measurer changes from state *a*0 to *a*1 at some instance and from a different state *a*0′ to *a*2 at another instance, where *a*0′ is not the same as *a*0, these changes cannot produce *symbols representing different external states* based on the principle of inverse causality. Therefore, autopoietic semiotic systems must be equipped with molecular devices that function as measurers with at least one baseline, which must return to the baseline after a state change. In particular, the ability to return to a baseline after changes is critical for IC measurements. 

Present-day LSs, including unicellular prokaryotes such as bacteria, have experienced approximately 3.8 billion years of evolution. Therefore, the LSs in the primitive and intermediate evolutionary stages of the semiotic cognitive system are missing. However, we can depict a possible scenario for several key evolutionary stages of semiotic cognition in terms of IC measurements. Hypothetically, during the unicellular evolution of LSs, an autopoietic metabolic system (Figure 7C) evolved by incorporating a chemical network of IC measurement systems from a primitive stage (i.e., *A^M^*-*D^M^*), in which *B^M^* functions as an effector, such as membrane transport proteins, through an intermediate stage (i.e., *A^M^*-*B^M^*-*D^M^*), to a developed stage (i.e., *A^M^*-*B^M^*-*C^M^*-*D^M^*).

Cell surface receptor proteins are considered to function as *A^M^*, which are composed of a receiver (or discriminator) domain that recognizes external stimuli, such as binding to ligand molecules or physical state changes. Binding is a highly specific, noncovalent interaction based on structural complementarity (the lock-key principle) and a transmitter (or effector) domain. The receptor protein exists in an allosteric equilibrium between inactive and active conformations. Ligand binding stabilizes either active (agonistic) or inactive (antagonistic) forms. Receptor proteins are coupled with ion channels, enzymes (kinases or phosphatases), or G proteins in contemporary cells, including prokaryotes and eukaryotes, sometimes called “molecular switches” [113]. The coupled molecular switch changes (flips) the state according to changes in the receptor state. These molecular switches are considered to function as *B^M^*, which measure (cognize) receptor state changes.

According to the IC measurement process, the state changes of receptor proteins as *A^M^* are “given” to the cell. In the above description, receptor states (active or inactive) are caused by physical and chemical stimuli from the perspective of an external observer (biologist). However, the stimuli are hidden entities external to the focal cell. Unlike external observers, a cell can only produce internal states that signify these entities’ states from the receptor states (i.e., data for the cell, not for an external observer who observes the cell and its surroundings). Receptor states are measured by another component of the cell, receptor-coupled switch proteins, which correspond to the *B^M^*. The states of a switch protein signify external states, such as the states of the ligands. The states of switch proteins can then be measured, mediated through measurement by *C^M^* (interpreter) or directly by *D^M^* (effector). 

A well-studied signal transduction of bacterial chemotaxis [114] provides a useful example of how an *A^M^*-*B^M^*-*D^M^* system (without *C^M^*) operates. Here, *A^M^* corresponds to a receptor protein (MCP, methyl-accepting chemotaxis protein); *B^M^* corresponds to a switch protein (CheA, a histidine kinase) and the carrier of the switch protein states (CheY), both with activated (phosphorylated) and non-activated states; *D^M^* corresponds to the flagellar motor with two states (modes), rotating counterclockwise (forward run) or clockwise (tumble), which has a switch protein FliM. Note that CheA is physically coupled with the receptor protein (MCP) through another protein (CheW); therefore, CheY functions as the second *B^M^*, which travels to transmit CheA states (activated or not) to the flagellar motor (*D^M^*). Thus, the second *B^M^* states are copies of the first *B^M^*. (We can also interpret that {MCP + CheW + CheA} corresponds to *A^M^* and CheY to *B^M^*). Furthermore, CheR methylates the MCP receptor, which is desensitized to its ligands. This modification of the MCP conformation can change the current baseline (*a*0) to another (i.e., baseline shift [16]), through which physiological adaptation occurs at a given stimulus concentration. This desensitization allows a cell to adapt to its current ligand concentration, where the tumbling mode of CheA activation is transmitted to motor rotation, causing it to swim in random directions. Another protein, CheB, in a phosphorylated (activated) state, can remove methyl from methylated MCPs. In the chemotaxis system, the degree of methylation is a type of memory stored in the molecular state of the receptor [113]. Using a methylation system involving the baseline shift, bacterial cells can detect *changes in the stimulus concentration* (not the concentration itself) in their environments. 

Similarly, the lac operon described in Appendix C is an example of an IC measurement process involving *A^M^*-*B^M^*-*D^M^* coupling. In this process, the repressor protein acts as an *A^M^*, changing its molecular conformation, leading to the gene expression of ß-galactosidase and permease as an adaptive response.

The *A^M^*-*B^M^*-*C^M^*-*D^M^* system operates in quorum sensing in bacteria. We can explain that *A^M^*-*B^M^* coupling measures “autoinducer concentrations”. *C^M^* states are not copies of *B^M^*; instead, they signify the “number of other members” that produce or cause autoinducer concentrations in the environment [16]. 

In the above examples of bacterial adaptation, we can take either an external or an internal perspective on how a bacterial cell behaves under given surrounding conditions, such as the presence of sugars and other bacterial cells. From an internal perspective, no entity outside a focal cell is assumed; that is, from a bacterial point of view. Here, the cell performs an IC measurement operation: given internal states of a system component (*A^M^*; note that “receptor” is an externalist language), it produces internal states of components (symbols or signals) that signify the hidden external states of nutrient or harmful molecules and processes them to determine how to act (swim). This process can operate as long as inverse causality operates correctly (described in Section 7.2).

Finally, it may appear that the solipsism problem (Appendix A) does not exist in LSs, including humans. This may suggest that they have solved this problem by developing IC operation systems during their evolution.

## 9. Conclusions

In this study, based on the conceptual extension of cognition in the CS model, the adaptation of LSs involves cognition at all three levels of the nested hierarchy of LS organization: (i) physical cognition: state changes through forces (e.g., gravity and electromagnetic fields); (ii) chemical cognition: state changes through chemical reactions; and (iii) semiotic cognition: state changes through the operation of inverse causality. Detection capabilities of physical and chemical entities (cognizers) can generate a higher level of cognition, i.e., semiotic cognition, if they form an IC operation system within an LS, wherein the system produces internal states as symbols that signify external states hidden from the subject LS and process them to act adaptively. 

Mind and matter can be understood within the monistic framework of the CS model. The mind involves the controversial concept of consciousness (although its existence does not need proof because it is directly experienced). Consciousness encompasses various features of the mind, such as sentience, awareness, intentionality, and attention, which are difficult to address using scientific methodologies. Therefore, we defined the mind in a tractable form using the cognizer concept, which is the process of generating symbols that signify external states, leading to actions (i.e., effecter state changes). This process is essentially semiotic cognition through inverse causality operation. LSs have invented semiotic cognitions based on physical and chemical cognitions to manage the probability distribution of events that occur to the cognizer. Recognizing consciousness as a process involving semiosis is not a novel concept explored in this study. Philosophers and semioticians have long sought to comprehend consciousness as a semiotic process, and Peirce’s semiotics is a significant contributor to this understanding [114,115,116].

This study proposes that semiotic cognition is an adaptive process, wherein inverse causality operations produce particular internal states as symbols that signify hidden external states. It also proposes that this operation makes LSs aware of the external world. This claim does not imply that any LS has a form of mind similar to that of humans. Hypothetically, the property of awareness or consciousness may vary according to evolutionary stages, ecological conditions, and social structures, resulting in various IC operation systems with different degrees of complexity in their structure. However, further investigation using biological data is required from this perspective. 

## Figures and Tables

**Table 1 entropy-26-00660-t001:** Four types of probability, entropy, and amount of information.

	Internal (Cognizers within a System)	External (External Observers)
Probability	Pcog, Poverall	Pcog, Poverall
Entropy	Hcog, Hoverall	Hcog, Hoverall
Amount of Information	Icog, Ioverall	Icog, Ioverall

**Figure 1 entropy-26-00660-f001:**
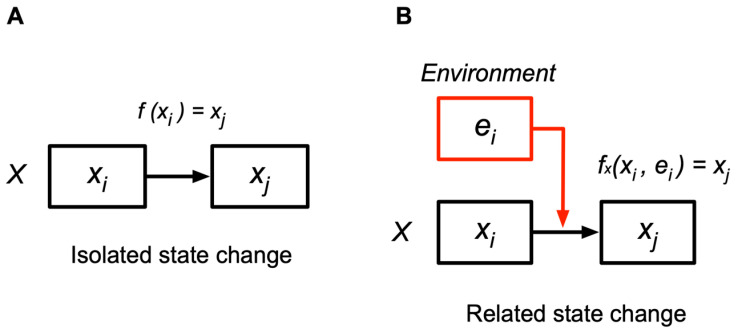
(**A**) Isolated state change refers to a change that occurs independently of the environment. (**B**) Related state change refers to a change that occurs depending on the environment, called cognition.

**Figure 2 entropy-26-00660-f002:**
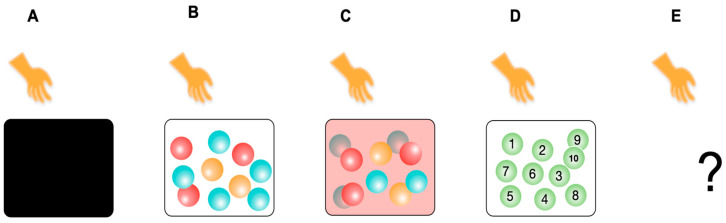
A person draws one ball from the box. What is the probability of drawing a particular ball? (**A**) The box is opaque and contains ten balls: two orange, three red, and five blue balls. (**B**) The box is transparent but otherwise the same as (**A**). (**C**) The box is semitransparent but otherwise the same as (**A**). (**D**) The box is transparent and contains ten balls numbered from 1 to 10. (**E**) The conditions are unknown from the player’s perspective.

**Figure 3 entropy-26-00660-f003:**
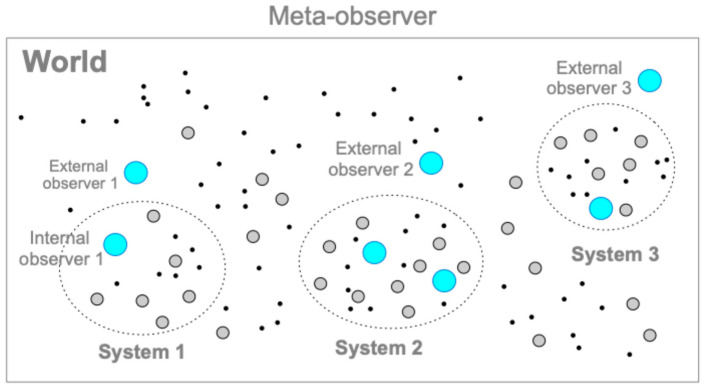
Framework of the cognizers system (CS) model (see text for differences between “cognizers system” and “cognizer system”). The meta-observer describes the world, which is deterministic. The world comprises cognizers. A system of cognizers is referred to as a cognizers system. Dots and gray circles indicate cognizers that perform physical and chemical cognitions, respectively. Internal and external observers are cognizers (living systems, not restricted to humans) who perform semiotic cognitions, denoted by blue circles. Systems 1, 2, and 3 show subsets of cognizers that the meta-observer can demarcate as a “system”.

**Figure 4 entropy-26-00660-f004:**
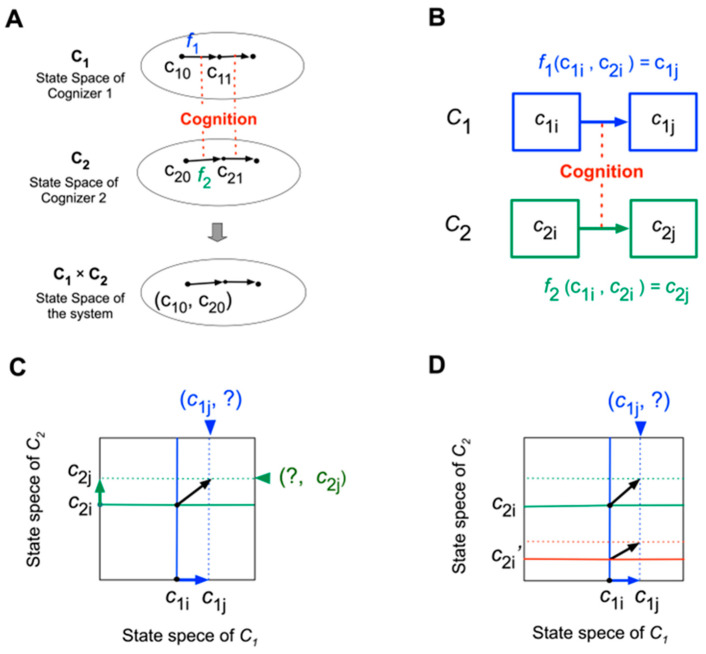
(**A**) Cognizers system comprising only two cognizers, *C*1 and *C*2, in the world. (**B**) Related state changes in *C*1 and *C*2 through cognition. (**C**) The determination (selection) of a succeeding state narrows down the relation to the other cognizer. (**D**) If *C*1 cannot discriminate between various states of *C*2, namely *c*21 and *c*2*i*′, then *C*1 will have uncertainty about the states of *C*2 that occur following the cognition *c*1*i* ⟼ *c*1*j*.

**Figure 5 entropy-26-00660-f005:**
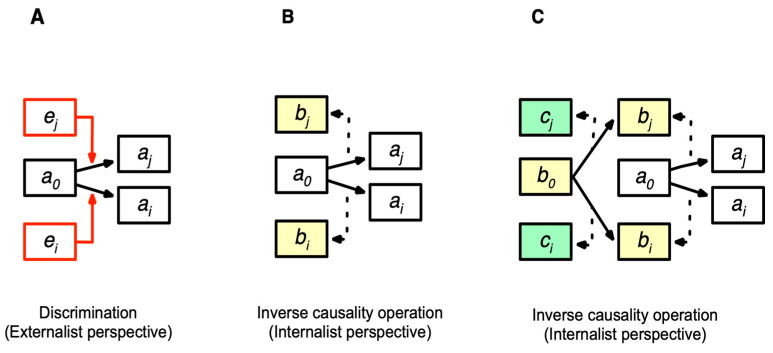
(**A**) From an external perspective, *A* in state *a*0 can discriminate between the external states. *ei* and *ej*. (**B**) From an internal perspective, external states, *ei* and *ej*, are hidden. Suppose that *A* has changed from *a*0 (a baseline state) to *ai* in some instances and from *a*0 to *aj* in others. In this case, different symbols, *bi* and *bj*, are introduced behind *a*0 (dotted arrows) to fulfil inverse causality (IC) by IC operation. The introduced symbols signify the hidden external states *ei* and *ej*, respectively. (**C**) *B* changes from a baseline state (*b*0) to *bi* or *bj*, which violates the principle of inverse causality. Therefore, different symbols, c*i* and c*j*, are introduced behind *b*0 by IC operation. The states of *C* (*ci* and *cj*) signify the hidden external states (e.g., *ek* and *el*, not represented in the figure) that *A* cannot detect.

**Figure 6 entropy-26-00660-f006:**
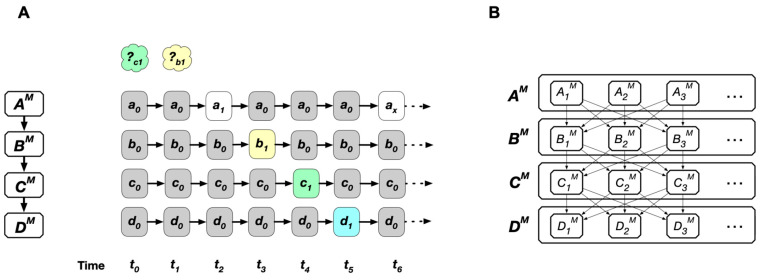
(**A**) IC operation system composed of measurers, *A^M^*, *B^M^*, *C^M^*, and *D^M^*. This diagram focuses on only a short process. The baseline states are denoted by the subscript “0”. The state changes from *t*0 to *t6* are shown; namely: (*a*0, *b*0, *c*0, *d*0), (*a*0, *b*0, *c*0, *d*0), (*a*1, *b*0, *c*0, *d*0), (*a*0, *b*1, *c*0, *d*0), (*a*0, *b*0, *c*1, *d*0), (*a*0, *b*0, *c*0, *d*1), (*a*0, *b*0, *c*0, *d*0) … The *A^M^* sequence includes *a*0 ⟼ *a*0 or *a*1, to which inverse causality is operated. This process transforms the backward-in-time IC algorithm (Figure 5B,C) into a measurement process forward in time, in which the distinctions made by each measurer are transmitted downstream in time. Modified from Figure 9 in [16]. (**B**) *A^M^*, *B^M^*, *C^M^*, and *D^M^* are composed of sub-measurers.

**Figure 7 entropy-26-00660-f007:**
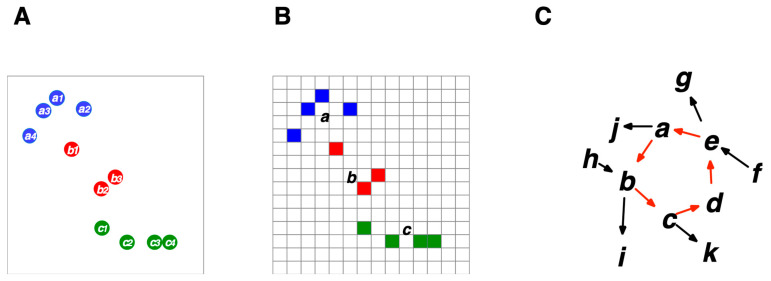
(**A**) The population of molecules described as particles that behave in the state space of position and velocity. The entire population includes subpopulations of type *a* molecules (*a*1, *a*2, …), type *b* (*b*1, *b*2, …), and type *c* (*c*1, *c*2, …). (**B**) A subpopulation of molecules of the same type is described as a field entity in which individual molecules (particles) are generated and degraded through chemical reactions. Molecules as fields (*a*, *b*, and *c*) behave in a density-state space. (**C**) Metabolic closure, which comprises molecular fields (*a*, *b*, *c*, *d*, and *e*, linked with red arrows), emerges as a molecular system in the density space. The molecular relationships in the chemical reactions are indicated by the red and black arrows; their spatial distributions, as in B, are not presented here.

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
