# Peer review of "Unification of Mind and Matter through Hierarchical Extension of Cognition: A New Framework for Adaptation of Living Systems"

_entropy, 2024, doi:10.3390/e26080660_

Round 1
Reviewer 1 Report (New Reviewer)
Comments and Suggestions for Authors# Evaluation
If I am right, the main meat of the paper comes in section 7. It would be good
if the main thrust of this section was introduced qualitatively earlier on. Much
of what is presented in the previous sections should also be rethought: does
this or that discussion help the reader prepare for §7? If not particularly, I
would consider it as a candidate for deletion, perhaps.
Regarding this part: is the author's main idea that a cognizer is
able to solve (what he sees as) the problem of telling hallucination from
veridical perception by:
1. Relying on the fact that each of the cognizer's states has one single
precursor, and
2. Using "inverse causality" to work out what that state was?
If it is, this seems to clash with the fact that we are absolutely not able to
do "inverse causality" reasoning in what appears to be the necessary precision
for the kind of result the author has in mind. Also, there is a huge literature
on the difference between hallucination and veridical perception. The author
should engage with it.
# Comments
p. 1:
> One might answer that organisms detect stimuli or information from their
surroundings and react to them appropriately. This is an externalist perspective
in which an external observer (scientist or experimenter) observes a given LS
and its surroundings from the outside. However, any LS cannot take this
perspective because it cannot observe itself and its surroundings from the
outside.
No: detecting information about one's surroundings does not require that one can
observe "[oneself] and [one's] surrounding from the outside". *Establishing that
this detection is going on* does require that, but this is irrelevant to your
current point, I think.
> How can passive, inert physical matter comprise an LS capable of achieving this?
Why should matter capable of achieving this be passive and inert?
I am not entirely seeing how this problem is related to the problem of
solipsism. I can see that there is a family resemblance here, but I don't think
that's enough for the kind of identification the author seems to be proposing.
For example, suppose that a living system solves the problem of distinguishing
hallucination from real perception (that is to say, it is able to distinguish
reliably enough when it is in one of these two situations) by, e.g., attending
at the multimodal agreement of its percepts: if it sees something but does not
hear it, this makes the subjective probability of hallucination higher, etc.
This can be done without solving, (indeed, without caring about) the problem of
solipsism.
p. 2:
> Symbol generation by inverse causality (called, the “IC measurement”) has been
> proposed as a universal principle for a system to be aware of the external
> world
Proposed by whom? And what is "symbol generation by inverse causality"?
p. 3:
> A question arises as to whether a group of organisms is conscious.
This observation comes a bit out of the blue here. I suggest that you first
discuss your main thesis (which we have only incompletely been introduced to so
far) for the simpler kinds of systems, and then move on to the possibility of
group consciousness.
> According to previous studies, we can characterize five properties of LSs
as material systems that distinguish them from nonliving entities.
You really need to cite those previous studies, and in general be more
forthcoming about where the ideas you are relying on come from.
Also, it's unclear to me where this very general discussion of life is going.
p. 5:
> Natural selection tests adaptations, not produce them
But it does, at least in the sense of making incremental changes that result in
better and better versions of adaptive traits. Cue the usual discussion of how
photosensitive tissue gives way to rudimentary, and then less and less
rudimentary, eyes.
> “Information” is reified as a third entity in addition to matter and energy
> without explicit definition.
Again, who does this? More citations would be very welcome. There are, of
course, fully explicit definitions of information in the Shannonian tradition.
> In the framework of CS model, information is defined as a “related (or
> relational) state change”
By whom, where?
p. 6:
> An isolated state change from x to y is represented as xi ⟼ xj
You mean from xi to xj, right?
> The CS mode describes it as your hand cognizing the stone and the stone
> cognizing your hand (changing velocity and position).
OK, but what do we gain by redefining cognition in such a trivializing way?
p. 13:
> Most researchers have not made it explicit in the literature whether the
> probability “P( )” is for an internal observer (cognizer) in interaction with
> the object events (i.e., internal probability), or an external observer, such
> as an experimenter (i.e., external probability).
Examples of researchers making this mistake would be welcome.
p. 18:
> Any cognizer cognizes all other cognizers wherever they exist.
What if some cognizers probabilistically screen others off? Also, isn't the
"principle of local causation" in §6.7 incompatible with the above claim?
p. 21:
> From the internalist framework (Section 5: Experiment E), the environment 1052
for A is not assumed (but not denied), as in Husserl’s phenomenology [74,75].
What does this sentence mean?
The English is very good but would benefit from some light editing
Author Response
I attached a pdf file to reply the comments.

Reviewer 2 Report (New Reviewer)
Comments and Suggestions for Authors
Please see my extensive attached pdf file, which treats many parts of the paper in exhaustive detail.

My comments are to be found entirely in the attached pdf file.
Author Response
I attached a pdf file to reply to the comments.

Round 2
Reviewer 2 Report (New Reviewer)
Comments and Suggestions for Authors
I cannot recommend publication in the MS's current form, too many things require clarifying. I have therefore made extensive Comments in the Margin.
Many make required corrections in the English, which may surprise you as the English is basically extremely good; but, for all the mistakes or poor quality expressions that I noticed, I have made corrections or constructive suggestions.
I made a number of quite severe scientific comments; mostly because your understanding of the details of how quantum theory works seems inadequate. I could not allow an MS to be published without correcting those mistakes. (Part of my PhD research at MIT was in quantum field theory and elementary particle physics, so I assure you that I am qualified to help you in this way.)
I also made some severe comments on how you use the terms, cognizer and cognition. But I do realise that you have developed a well thought through personal perspective on this over a number of years, and that it is at the heart of what you are communicating in this lengthy series of articles (Refs 7 to17), this being the latest. And, by the way, thank you for adding references 1 - 6, especially nos 5 & 6 to Brian Josephson's work.
Please especially note my comment where I stated that I had had real difficulty in following your various definitions, and that you might consider improving them. As I also state, I am not convinced of the validity of your perspective, but I am still going to recommend that, after you make the required / suggested changes that your MS be published in Entropy.
However, I do hope that you will withdraw the current Preprint version of this article, which I happened to find on Google Scholar(!), and that you will only make another one available, when the suggested corrections have been incorporated, and the article is definitely accepted for publication.
Finally, a word of explanation: Entropy leaned on me to submit my first review at a time when I was exceptionally busy / preoccupied. I made very limited suggestions and offered to re-review your MS. This second attempt can be taken as my real review.

Author Response
Please see attachment

This manuscript is a resubmission of an earlier submission. The following is a list of the peer review reports and author responses from that submission.
Round 1
Reviewer 1 Report
Comments and Suggestions for Authors
The manuscript "Unification of mind and matter through hierarchical extension of cognition: A new framework for adaptation of living systems", by Toshiyuki Nakajima, is a semiotic/philosophical exercise that elegantly "solves" the Cartesian schism on cogitation: cognition is extended to all layers of physical-chemical realities. Mind and matter are unified as cognizers, and the principle of inverse causality takes care of awareness of the external world. Four centuries of debate are evaporated. Great. Well, this reviewer, working in different facets of biological information, was initially interested in the proposals of the manuscript, but was disappointed to find that all the work was finally a semiotic/philosophical discourse devoid of scientific interest per se. Thus, given that the thematic lens of this reviewer is not focused on the philosophical debate, and not finding sufficient elements of scientific interest—the present revision returns empty... It is a pity because in some points (bonds, molecular recognition, evolutionary comments) the author shows an interesting background. But the lengthy paper, overall, does not look suitable for scientific, multidisciplinary reading.
Reviewer 2 Report
Comments and Suggestions for Authors
The author presents a thesis in which "cognition" is defined as the "perception of the external world". This thesis is incorrect and contradicts the fact that external/internal is system dependent.
As an example consider a single organism. With respect to the organism, the external world is the environment. Perception of the environment such as "vision" is necessary and one aspect of "cognition". However, the organism also has a perception of its internal condition, e.g. "pain" produced by biochemical molecules (prostaglandins and leukotrienes) inside the organism body, which is also "cognition".
The main focus only on "external perception" as a basis of cognition is wrong and the division of internal vs. external is not fundamental. Organism is composed of organs, then the organs are made of tissues, the tisues are made of cells, the cells are made of organelles, the organelles are made of biomolecules. Going only in one direction as the author defines "cognition" only in the case of "external perception" means that an organism can never perceive the functioning of its organs, and organ can never perceive the functioning of its tissues, the tissue can never perceive the functioning of its cells, and so on.
This work needs to answer the fundamental question whether smaller "minds" can exist inside bigger "minds". If the answer is YES, then this would imply that a group of people can have a collective "mind" and the universe as a whole has a "universal mind". Is the author ready to defend the existence of a "universal mind"? If the answer is YES, then start the article with the discussion on the "universal mind" so that the reader clearly knows whether to read further or not without wasting precious time.
Comments on the Quality of English LanguageEnglish is OK.
Reviewer 3 Report
Comments and Suggestions for Authors
Please find my comments in the attached file.
I would recommend the author to make a connection to the cognitive science and currently dominant view of cognition as well as to the new work on basal cognition.
